# BH3 mimetics targeting BCL-XL have efficacy in solid tumors with RB1 loss and replication stress

Andreas Varkaris[1,2,8], Keshan Wang[1,3,8], Mannan Nouri[1,8], Nina Kozlova[1,8], Daniel R. Schmidt[1,4], Anastasia Stavridi[1], Seiji Arai[1], Nicholas Ambrosio[1], Larysa Poluben[1], Juan M. Jimenez-Vacas[5], Daniel Westaby[5], Juliet Carmichael[5], Fang Xie[1], Ines Figueiredo[5], Lorenzo Buroni[5], Antje Neeb[5], Bora Gurel[5], Nicholas Chevalier[2], Lisha Brown[6], Olga Voznesensky[1], Shao-Yong Chen[1], Joshua W. Russo[1], Xin Yuan[1], Dejan Juric[2], Himisha Beltran[7], Johann S. De Bono[5], Matthew G. Vander Heiden[4,7], David J. Einstein[1], Taru Muranen[1], Eva Corey[6], Adam Sharp[5] & Steven P. Balk[1] ✉

BH3 mimetic drugs that inhibit BCL-2, BCL-XL, or MCL-1 have limited activity in solid tumors. Through assessment of xenograft-derived 3D prostate cancer models and cell lines we find that tumors with *RB1* loss are sensitive to BCL-XL inhibition. In parallel, drug screening demonstrates that disruption of nucleotide pools by agents including thymidylate synthase inhibitors sensitizes to BCL-XL inhibition, together indicating that replication stress increases dependence on BCL-XL. Mechanistically we establish that replication stress sensitizes to BCL-XL inhibition through *TP53/CDKN1A*-dependent suppression of *BIRC5* expression. Therapy with a BCL-2/BCL-XL inhibitor (navitoclax) in combination with thymidylate synthase inhibitors (raltitrexed or capecitabine) causes marked and prolonged tumor regression in prostate and breast cancer xenograft models. These findings indicate that BCL-XL inhibitors may be effective as single agents in a subset of solid tumors with *RB1* loss, and that pharmacological induction of replication stress may be a broadly applicable approach for sensitizing to BCL-XL inhibitors.

The anti-apoptotic BCL-2 family proteins (primarily BCL-2, BCL-XL, and MCL-1) act by neutralizing BAX and BAK, and by inhibiting the BH3-only pro-apoptotic proteins that can activate BAX/BAK[1]. These interactions are mediated by the BH3 domain, and BH3-mimetic drugs can enhance apoptosis by binding to and inhibiting BCL-2, BCL-XL, or MCL-1. ABT-737[2] and ABT-263 (navitoclax, orally bioavailable analog of ABT-737)[3] are BH3-mimetics that neutralize BCL-2, BCL-XL, and BCLW[4]. Navitoclax has single-agent activity in hematological malignancies[5], but causes thrombocytopenia due to on-target effects on megakaryocytes. A BCL-2-specific agent that spares platelets (ABT-199, venetoclax) is similarly active and FDA approved for several hematological malignancies[6,7].

[1]Department of Medicine, Beth Israel Deaconess Medical Center, Harvard Medical School, Boston, MA, USA. [2]Massachusetts General Cancer Center and Department of Medicine, Harvard Medical School, Boston, MA, USA. [3]Department of Urology, Union hospital, Tongji Medical College, Huazhong University of Science and Technology, Wuhan, China. [4]Koch Institute for Integrative Cancer Research, Massachusetts Institute of Technology, Cambridge, MA, USA. [5]Institute of Cancer Research, Royal Marsden NHS Foundation Trust, London, UK. [6]Department of Urology, University of Washington, Seattle, WA, USA. [7]Dana-Farber Cancer Institute, Boston, MA, USA. [8]These authors contributed equally: Andreas Varkaris, Keshan Wang, Mannan Nouri, Nina Kozlova. ✉e-mail: sbalk@bidmc.harvard.edu

In contrast to hematological malignancies, these currently clinically available BH3-mimetics targeting BCL-2 and BCL-XL have limited single-agent activity in solid tumors[8,9]. One basis for this relative resistance is ineffective blockade of MCL-1, which is substantially expressed in most solid tumors, and high levels of which are associated with resistance[8,10–14]. Indeed, we reported that down-regulation of MCL-1 by RNAi or CRISPR can dramatically enhance the apoptotic response to navitoclax in prostate cancer cells in vitro and in vivo, with complete responses in established prostate cancer xenografts[15]. The efficacy of dual BCL-XL and MCL-1 targeting has also been shown in multiple additional solid tumor models[16]. Moreover, navitoclax (or other drugs targeting BCL-XL) may be efficacious when used in combination with agents acting through mechanisms that include decreasing MCL-1[8,14,17–23] or increasing proapoptotic BH3-only proteins[24–26]. Finally, while the initially available BH3 mimetics targeted BCL-2 and BCL-XL, BH3 mimetics that target MCL-1 (such as

**Fig. 1 | Subset of PCa tumors is responsive to single-agent BCL-XL inhibition.**
**A** BH3 mimetic screening in PCa 2D and 3D models. Growth inhibition of PCa
models treated for 4 days with single agent Navitoclax (BCL-XL inhibitor), Vene-
toclax (BCL-2 inhibitor), or S63845 (MCL-1 inhibitor) over a dose range was assessed
compared to control. The fraction of models that could be inhibited by >50% for
each treatment at the highest concentration (1 µM) is shown. Left panel created in
Biorender, Yuan, X. (2025) https://BioRender.com/r54j854 **B** Heat map based on
ranking of IC50 values showing distribution of PCa model sensitivity to BH3
mimetics. **C** BIDPC5 spheroids were treated with navitoclax and S63845 for 6 h.
Apoptosis was assessed with luminescence-based caspase 3 and 7 activity assay.
Mean and SEM for 5 biological replicates are shown. Data were analyzed by one-way
ANOVA ***$p < 0.001$. **D** BIDPC5 (navitoclax sensitive) and BIDPC4 (navitoclax
resistant) spheroids were treated with navitoclax (1 µM) for 6 h. Apoptosis markers

cleaved PARP and cleaved caspase 3 (CC3) were assessed with immunoblotting.
**E** BIDPC5 spheroids were treated with navitoclax (1 µM) for 6 h. Apoptosis was
assessed with fluorescence-based caspase 3 and 7 activity assay. Scale bar is 100 µM
(**F**) BIDPC1 and BIDPC5 patient derived xenografts were treated with intraperitoneal
DMSO or navitoclax (50 mg/kg every other day) for 14 days. Fold change in tumor
volume was assessed after completion to therapy. Asterisk (*) represents mice that
developed toxicity requiring euthanasia during the 14 day treatment period. Data
were analyzed by unpaired $t$-test ***$p < 0.001$ (for PC1 $p = 0.0001$ and for PC5
$p = 0.0008$). Upper panel was created in Biorender. Yuan, X. (2025) https://
BioRender.com/f87k274 **G** Kaplan-Meyer curves show overall survival. Data were
analyzed by logrank test. Data are presented as mean values +/- SEM. Source data
are provided as a Source Data file.

S63845, AZD5991, and AMG-176) are now becoming available and
entering clinical trials[16,24,27–29]. These agents in combination with
navitoclax can induce rapid apoptotic responses, but toxicity will
likely limit the ability to combine these agents in the clinic[30,31].
Together these findings indicate that BH3 mimetics targeting BCL-XL
may be effective as single agents in a subset of solid tumors with
particular genomic alterations that increase dependence on BCL-XL,
or in combination with other agents that have effects on apoptosis
pathways.

In this work we screen a diverse panel of prostate cancer patient-
derived models for responses to navitoclax, and find that *RB1* loss is
associated with increased sensitivity. In parallel, in a comprehensive
drug screen for agents that sensitize to navitoclax we identify drugs
including thymidylate synthase inhibitors that disrupt deoxyr-
ibonucleotide pools, together indicating that replication stress sensi-
tizes to BCL-XL inhibition. We further show that replication stress
renders tumor cells responsive to BCL-XL inhibitors, determine the
mechanistic basis for this response, and show that it can be leveraged
in vivo for effective therapy.

## Results

### Subset of prostate cancers are responsive to single agent BH3 mimetic agents

We initially assessed responses to navitoclax (targeting BCL-2/BCL-XL),
venetoclax (targeting BCL-2), and S63845 (targeting MCL-1) in a series
of prostate cancer PDX-derived primary cultures/3D spheroids,
patient-derived organoids, and cell lines (Fig. 1a, Supplementary
Table S1). Navitoclax at the maximal dose used was able to suppress
cell recovery by at least 50% in a subset of the PDX derived 2D and 3D
cultures, with the lowest IC50 values in 3D cultures (spheroids) from
BIDPC1 (125 nM) and BIDPC5 (100 nM) (Fig. 1b). Venetoclax had mini-
mal activity, consistent with the response to navitoclax being primarily
through BCL-XL. Moreover, with the exception of BIDPC6, tumors that
were responsive to navitoclax had minimal responses to S63845, fur-
ther indicating that BCL-XL is playing a dominant role in suppressing
apoptosis in this subset of tumors. Notably, S63845 (but not navito-
clax) was highly effective in the VCaP cell line and in 3D cultures from
the LuCaP35CR and 70CR PDXs, indicating a dominant role for MCL-1
in these cells.

The efficacy of navitoclax in the BIDPC1 and BIDPC5 models was
further assessed in additional cultures carried out to 7 days (Supple-
mentary Fig. S1a). The marked decreases in cell recovery in the
navitoclax-responsive models was consistent with apoptosis. This was
confirmed by examining caspase activity in BIDPC5 cultures exposed
to navitoclax, which showed rapid induction of caspase activity based
on cleavage of a caspase 3/7 substrate (Fig. 1c) and immunoblotting for
cleaved PARP and cleaved caspase 3 (Fig. 1d). Notably, while S63845
alone was not effective, it did enhance the response to navitoclax,
indicating that MCL-1 contributes to suppressing apoptosis in these
cells, but that BCL-XL has a more dominant role (Fig. 1c). To assess
whether apoptosis was being induced in a large fraction of the cells, we

carried out imaging with a fluorescent agent that labels apoptotic cells.
This analysis confirmed that navitoclax was driving apoptosis in the
majority of cells (Fig. 1e).

To confirm that navitoclax sensitivity was not related to
experimental conditions used in vitro (culture media, cell attach-
ment, proliferation rate) we examined the BIDPC1 and BIDPC5 PDXs
for responses to navitoclax in vivo. Subcutaneous PDXs were estab-
lished in the flanks of immunodeficient mice, and we confirmed RB1
loss in the PDXs by IHC (Supplementary Fig. S1b). Navitoclax treat-
ment was then initiated when tumors reached ~500 mm³. Notably,
there was marked regression in all navitoclax treated tumors, with
complete responses in two of the BIDPC1 mice (Fig. 1f). Moreover,
there was a significant survival advantage for the navitoclax treated
mice (Fig. 1g). Together these findings indicate that agents targeting
BCL-XL may be effective as single agents in a subset of prostate
cancers.

### Tumors with loss of *RB1* activity have increased dependency on BCL-XL

Examination of genomic alterations in the navitoclax-responsive
tumors showed an association with loss of *RB1* function. BIDPC1 and
BIDPC5 had losses of *RB1* and *BRCA2* (which are adjacent on chromo-
some 13 and frequently lost together in prostate cancer)[32] (Fig. 2a). The
*RB1* loss was also confirmed by IHC (Supplementary Fig. S1b).
LuCaP176 had loss of *RB1* and *TP53*, and BIDPC6 had loss of *CDKN2A/2B*
(which phenocopies loss of *RB1*). Together these observations sug-
gested that tumors with loss of *RB1* function may have an increased
dependence of BCL-XL. To further assess this association, we inde-
pendently examined the efficacy of navitoclax inhibition in two addi-
tional *RB1* null PDXs (which were wild type for *BRCA2*) and two similarly
generated *RB1* intact PDXs. Navitoclax treatment resulted in apoptotic
responses and decreased cell recovery in organoid cultures generated
from both of the *RB1* null PDXs (Fig. 2b).

We next used the Genomics of Drug Sensitivity in cancer database
(Sanger Institute and Mass General Cancer Center, https://www.
cancerrxgene.org/) to assess the effects of *RB1* alterations in multiple
solid tumor derived cell lines on the IC50 values for a large series of
drugs[33]. Notably, IC50 values for treatments with navitoclax, WEHI
(BCL-XL inhibitor), and ABT737 (BCL-2/BCL-XL inhibitor) were sig-
nificantly lower in cells with *RB1* alterations (combined mutation or
loss) (Fig. 2c), and were also lower when the analysis was just for copy
number loss (Supplementary Fig. S2). Moreover, amongst all drugs
tested, *RB1* alterations most significantly increased sensitivity to navi-
toclax (Fig. 2d). Conversely, as previously described, *RB1* mutations
markedly increased resistance to the CDK4/6 inhibitor palbociclib. The
IC50s for navitoclax in tumors with *BRCA2* or *BRCA1* loss were also
decreased, but with much lower effect sizes (−0.0643 and −0.0717,
respectively) and were not significant (Supplementary Fig. S3a), indi-
cating that *RB1* loss is the major driver of navitoclax-sensitization in the
tumors with combined *RB1* and *BRCA2* loss. Consistent with this find-
ing, the *BRCA2* deficient PACAN1 cell line was not sensitive to

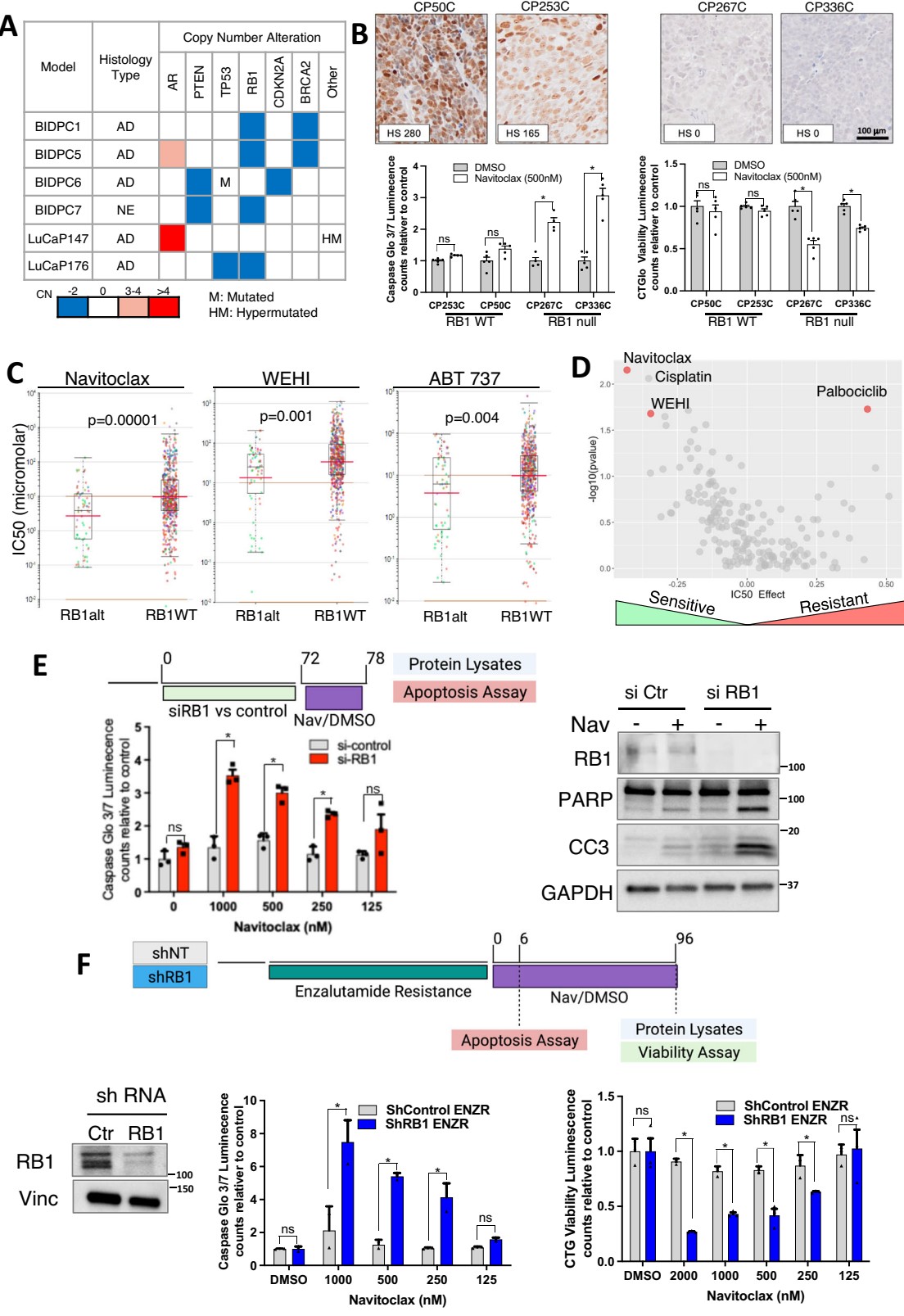

navitoclax (Supplementary Fig. S3b). Nonetheless, the potency of navitoclax in the BIDPC1 and BIDPC5 tumors suggests that *BRCA2* loss may contribute to sensitivity.

We next used RB1 siRNA to determine the effect of RB1 down-regulation on the response to navitoclax. LNCaP prostate cancer cells (*RB1* intact) were treated with RB1 or control siRNA for 3 days, and then treated with navitoclax for 6 h. Measurements of caspase activity

showed that RB1 siRNA sensitized to navitoclax (Fig. 2e, left panel). The siRNA-mediated RB1 depletion, and increases in PARP1 cleavage and cleaved caspase 3, were also confirmed by immunoblotting (Fig. 2e, right panel). In parallel, we examined LNCaP cells stably expressing an RB1 shRNA. Notably, RB1 protein in these cells when grown in complete medium was not substantially decreased. However, growth in medium containing the androgen receptor (AR) antagonist enzalutamide

**Fig. 2 | RB1 loss increases dependency on BCL-XL. A** Genomic characterization of PCa models based on PCa-commonly altered genes. **B** Validation of RB1-loss induced sensitivity to BCL-XL inhibitor using an independent PCa PDX-derived spheroid cohort. Upper panels: RB1 expression by IHC and H-scores. Lower panels: organoid cultures from each PDX were treated with navitoclax for 6 h and caspase activity was assessed (left), or treated for 4 days and cells viability was assessed (right). Mean and SEM for 5 biological replicates are shown. Data were analyzed by one-way ANOVA *$p < 0.05$ (left panel: $p = 0.003$ for model CP267C and $p = 0.002$ for model CP336C, right panel: $p = 0.002$ for model CP267C and $p = 0.001$ for model CP336C). Scale bar is 100 µM. **C** Comprehensive analysis of solid tumor cell lines sensitivity to BCL-XL inhibitors (Navitoclax, WEHI and ABT737) based on *RB1* alteration (combined mutation or copy number loss). Data were analyzed by unpaired *t*-test. **D** Volcano plot with effect size (*x* axis) and significance (*y* axis) of large-effect cancer-specific pharmacogenomic interactions based on RB1 alteration. Each circle represents an association between RB1 status and drug sensitivity analyzed using ANOVA (Genomics of Drug Sensitivity in Cancer- Sanger Institute/ Mass General Cancer Center database). **E** Effect of short term RB1 silencing on BCL-XL sensitivity in LNCaP (RB1 proficient PCa cell line). Cells were treated with siRNA

targeting RB1 (siRB1) or nontarget control (siNT) for 48 h. Navitoclax was then added to media for 6 h. Apoptotic effect was assessed by luminescence assay over a range of navitoclax concentrations (left panel) and apoptosis markers at 500 nM navitoclax by immunoblotting (right panel). Mean and SEM for 3 biological replicates are shown. Effects of RB1 siRNA at each navitoclax concentration were analyzed by unpaired *t*-test, *$p < 0.05$. Two-way ANOVA then showed that the effect of the shRNA on response to navitoclax was significant ($p = 0.002$). **F** Effect of long-term RB1 silencing on BCL-XL sensitivity in LNCaP cells. Cells were infected with shRNA targeting RB1 (shRB1) or nontarget control (shNT) constructs and treated with increasing doses of enzalutamide until development of resistance to 5 µM. Left panel: RB1 expression in the enzalutamide adapted cells. Middle panel: apoptosis activity of cells treated with navitoclax for 6 h based on luminescence assay. Right panel: viability assay of cells treated with navitoclax for 4 days based on luminescence assay. Mean and SEM for 3 biological replicates are shown. Effects of RB1 shRNA at each Navitoclax concentration were analyzed by unpaired *t*-test, *$p < 0.05$. Two-way ANOVA then showed that the effect of the shRB1 on response to Navitoclax was significant ($p = 0.0001$ for both apoptosis and viability analysis). Source data are provided as a Source Data file.

---

selected for markedly decreased RB1 protein in cells with the shRB1 compared to control cells similarly cultured in enzalutamide (Fig. 2f, left panel). This is consistent with previous data showing that AR inhibition causes cells to arrest in G0/G1, and this block can be at least partially overcome by RB1 loss[34]. Therefore, we examined the LNCaP cells expressing RB1 or control shRNA that were adapted to culture in 10 µM enzalutamide (enzalutamide-resistant, ENZR). Treatment with navitoclax greatly increased caspase activation in the RB1 shRNA cells compared to the control cells (Fig. 2f, middle panel). Similarly, navitoclax markedly reduced cell recovery in the RB1 shRNA cells, versus minimal effect in the control cells (Fig. 2f, right panel). Together these data indicate that RB1 loss increases tumor cell dependence on BCL-XL.

### Small molecule screen identifies agents that sensitize to navitoclax

In parallel with the above studies, we carried out a small molecule screen for agents that could sensitize prostate cancer cells to navitoclax. LNCaP cells were cultured in 384 well plates in the presence of navitoclax or DMSO for 2 days, followed by addition of agents in a mechanisms of action drug library containing ~1245 drugs applied at 4 concentrations for 2 days (see schema in Fig. 3a). Recovery of viable cells was then assessed by Cell Titer Glo assays. The screen identified multiple agents that enhanced the response to navitoclax, and these fell into several broad categories (Fig. 3b, Supplementary Table S2). Many were kinase inhibitors, which we showed previously could sensitize to navitoclax by increasing MCL-1 degradation through activation of an integrated stress response[15,22]. The next largest group were agents that impair the mitotic spindle leading to mitotic arrest, which also increases MCL-1 degradation[23]. The third largest group were CDK inhibitors that may also increase MCL-1 degradation or, through inhibition of CDK9, decrease MCL-1 transcription[35–37].

Amongst those in the "Other" category were agents that disrupt nucleotide pools, which can lead to replication stress and a DNA damage response. Notably, one consequence of *RB1* loss is replication stress, which may reflect both premature entry into S phase and a direct role for RB1 in DNA repair[38,39]. To determine whether a consequence of RB1 loss is replication stress and subsequent DNA damage response, we used doxycycline-regulated shRNA to suppress RB1 expression in two additional cell lines. RB1 downregulation in NCI-H2030 lung cancer cells was associated with an increase in phosphorylation of RPA32 (Supplementary Fig. S4a) and increased sensitivity to navitoclax (Supplementary Fig. S4b). RB1 downregulation in MCF7 breast cancer cells caused increased phosphorylation of both RPA32 and H2A.X (Supplementary Fig. S4a), although this was not

associated with increased navitoclax-mediated apoptosis (not shown). Consistent with these DNA damage response results, single sample gene set enrichment analysis (GSEA) of TCGA primary prostate cancers showed that those with *RB1* loss had an increase in the gene set ATR Activation in Response to Replication Stress (Fig. 3c). Moreover, *RB1* loss in prostate cancer is associated with increased expression of genes involved in DNA damage repair (Supplementary Fig. S4c).

### Replication stress increases dependence on BCL-XL

Based on these data, we next focused on whether, and by what mechanisms, replication stress and DNA damage increases dependence on BCL-XL. We initially used a thymidylate synthase inhibitor identified in the screen (nolatrexed) to selectively disrupt deoxynucleotide pools and cause replication stress, without direct effects on RNA synthesis. LNCaP cells were treated for 48 h with nolatrexed, which as expected increased the fraction of cells in S phase (Fig. 3d, left panel) and decreased cell recovery (Fig. 3d, middle panel). We then treated for an additional 6 h with navitoclax or vehicle control, which further decreased cell recovery in the nolatrexed treated cells, but not the control cells (Fig. 3d, middle panel). This decrease in cell recovery was associated with a marked increase in apoptosis, as assessed by caspase activation (Fig. 3d, right panel) and by immunoblotting for cleavage of PARP1 and caspase 3 (Fig. 3e). Finally, we treated with thymidine to confirm that the effects of nolatrexed were due to thymidylate synthase inhibition, and confirmed that this prevented the induction of apoptosis by navitoclax (Fig. 3f).

The disruption of deoxynucleotide pools causes replication fork stalling, and this replication stress can lead to increases in DNA single and double strand breaks with a subsequent DNA damage response and activation of ATR and ATM. To confirm that nolatrexed was causing replication stress we carried out DNA fiber assays. LNCaP cells were treated with nolatrexed for 16 h and then labeled with CldU followed by IdU. Nolatrexed treatment resulted in a highly significant decrease in both IdU track length and fork speed by ~50%, indicating that it was causing replication stress (Supplementary Fig. S5).

To determine whether this replication stress was causing a DNA damage response, we assessed for phosphorylation of ATR and ATM substrates, RPA32 and H2A.X, respectively. By immunoblotting we found that phosphorylation of RPA32 and H2A.X were increased by nolatrexed (Fig. 4a). Moreover, by COMET assays we determined that treatment with nolatrexed, and the related thymidylate synthase inhibitor raltitrexed, caused DNA damage that was comparable to that induced by doxorubicin (Fig. 4b). To confirm that this DNA damage response was due to thymidine depletion, we assessed the effects of adding thymidine to the medium. Indeed, thymidine supplementation

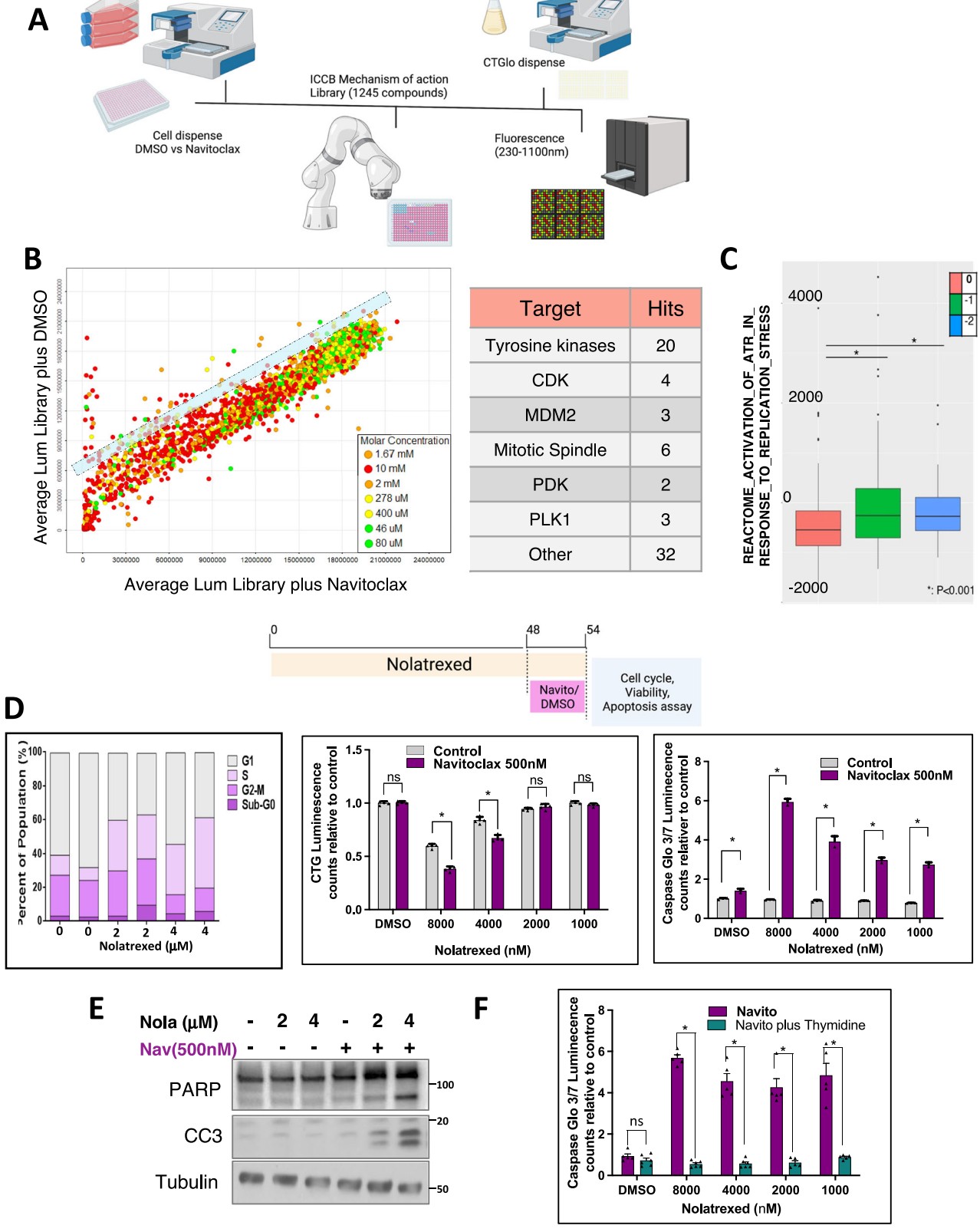

prevented the induction of a DNA damage response by nolatrexed and raltitrexed (Fig. 4c).

Consistent with the induction of DNA damage by raltitrexed, it also sensitized cells to navitoclax-mediated apoptosis (Fig. 4d). This sensitization was seen at 20 nM, further supporting an on-target effect, and was maximal at <100 nM (Supplementary Fig. S6a). We also carried out mass spectrometry to further confirm inhibition of thymidylate

synthase. Consistent with thymidylate synthase inhibition, raltitrexed (100 nM) for 24 h caused a dramatic depletion of dTTP that was associated with an increase in dUMP, the substrate for thymidylate synthase (Supplementary Fig. S6b). Comparable effects were seen with another thymidylate synthase inhibitor, pemetrexed.

The apoptotic response to navitoclax was also enhanced by other agents from our drug screen that cause replication stress, including

**Fig. 3 | Replication stress increases dependency on BCL-XL. A** Mechanism of action drug screening to identify agents that synergize with navitoclax. LNCaP cells were cultured in navitoclax (500 nM) or DMSO containing media. Compounds from the ICCB-Longwood Mechanism of Action Library (ICCB-L MoA) were then added in duplicate at 4 concentrations were then added, and viability assay was performed after 48 h. This figure was created in Biorender. Yuan, X. (2025) https://BioRender. com/p38v199 **B** Left panel: average luminescence (viability) for cells cultured with library drug alone (*Y* axis) versus with navitoclax (X-axis). Results for all 4 drug concentrations are plotted. Circles above blue lane represent potential positive hits. Right panel: drug class of positive hits. **C** Single sample gene set enrichment analysis of TCGA primary prostate cancer database was carried out. Activation of ATR Response to Replication Stress gene set was calculated for individual samples and plotted relative to RB1 copy number. Data were analyzed by one-way ANOVA *p < 0.05 (p = 0.002). **D** LNCaP cells were treated with nolatrexed or vehicle for 48 h followed by navitoclax or vehicle for 4 days for cell recovery or 6 h for apoptosis.

Left panel: cell cycle analysis of nolatrexed-treated LNCaP cells. Middle panel: cell recovery assessed by CTGlo assay. Right panel: caspase activation assessed by Caspase Glo 3/7 assay. Data are mean and SEM for biological replicates. Data at each nolatrexed concentration were analyzed by unpaired *t*-test, *p < 0.05. Two-way ANOVA then showed the effect of nolatrexed on response to navitoclax was significant (*p < 0.0001* for both apoptosis and viability analysis). Upper panel was created in Biorender. Yuan, X. (2025) https://BioRender.com/h91a470 **E** Analysis of apoptosis markers by immunoblotting of LNCaP cells treated with nolatrexed or vehicle for 2 days followed by navitoclax or vehicle for 6 h. **F** LNCaP cells were treated with nolatrexed alone or combined with thymidine for 2 days, followed by 6 h with navitoclax or vehicle, and apoptosis was assessed by CaspaseGlo 3/7 assay. Mean and SEM for 5 biological replicates are shown. Data at each nolatrexed concentration were analyzed by unpaired *t*-test, *p < 0.05. Two-way ANOVA then showed that the effect of adding thymidine was significant, *p < 0.001*. Source data are provided as a Source Data file.

BML-277 (CHK2 inhibitor) (Fig. 4e). We also confirmed that nolatrexed, as well as BML-277, sensitized to an independent BCL-2/BCL-XL inhibitor, AZD4320 (Fig. 4f, g). We then confirmed that thymidylate synthase inhibition sensitized to navitoclax in additional diverse cell lines including ZR75 (breast cancer), A549 (lung adenocarcinoma), A375P (melanoma), and RKO (colon cancer) (Fig. 4h). We also examined the effects of a BCL-2/BCL-XL proteolysis targeting chimera, PZ18753B. Treatment with PZ18753B markedly decreased BCL-XL, with a very modest effect on BCL-2 and no clear effect on MCL-1 (Fig. 4h), and induced apoptosis in cells treated with nolatrexed (Fig. 4i). Treatment with another thymidylate synthase inhibitor (5-fluorouracil, 5-FU) similarly sensitized to PZ18753B mediated apoptosis (Supplementary Fig. S7a). Finally, we examined a BCL-XL selective proteolysis targeting chimera (DT2216) (Supplementary Fig. S7b) and found that it similarly induced apoptosis in cells treated with raltitrexed (Supplementary Fig. S7c), further demonstrating that the effect is due to BCL-XL inhibition.

### Replication stress sensitization to navitoclax is mediated by decreased *BIRC5*/Survivin expression

We next focused on the molecular basis for sensitization to navitoclax in response to replication stress. Using siRNA we confirmed that apoptosis in response to nolatrexed plus navitoclax was decreased by depletion of BAX or BAK (Fig. 5a, b). However, nolatrexed did not increase expression of BAX or BAK. Notably, many agents can sensitize to BCL-XL inhibition by decreasing MCL-1 transcription or translation, or increasing its degradation. However, levels of MCL-1, or of BCL-XL or BCL-2, were not altered in response to nolatrexed (Fig. 5c). We next examined levels of the BH3 only proteins that can neutralize BCL-2/BCL-XL or MCL-1, and at varying degrees can directly activate BAK/BAX. We did not observe alterations in BIM, BID, BAD, NOXA, or MARCH5 (ubiquitin ligase targeting MCL-1)[22], but did find a substantial increase in PUMA (encoded by the *BBC3* gene) (Fig. 5d, e). Conversely, we found a marked decrease in the anti-apoptotic protein Survivin, encoded by the *BIRC5* gene (Fig. 5e).

To determine whether increased PUMA was sensitizing to navitoclax, we used siRNA to deplete PUMA and assessed apoptosis in response to nolatrexed combined with navitoclax. However, PUMA depletion did not decrease the apoptotic response (Supplementary Fig. S8). In contrast, Survivin depletion by siRNA sensitized cells to navitoclax (Fig. 5f, g), and this sensitization was not further enhanced by nolatrexed (Fig. 5f). Notably, Survivin depletion did not sensitize to S63845 (MCL-1 inhibitor), consistent with BCL-XL playing a dominant role in suppressing apoptosis in these cells (Fig. 5g). We next examined the effects of YM-155, a drug that decreases Survivin expression (although the molecular basis for this decrease is not yet clear) (Fig. 5h). We found that YM-155 markedly sensitized cells to navitoclax

(Fig. 5i), and similarly sensitized to another BCL-XL inhibitor, AZD4320 (Fig. 5j).

We similarly assessed effects of RB1 downregulation on expression of apoptosis related proteins. RB1 knockdown in MCF7 cells resulted in increased levels of MCL1, BIM, and BAX, while Survivin (which was expressed at low basal levels) was decreased (Supplementary Fig. S9). In contrast, Survivin was expressed at high basal levels in NCI-H2030 cells and was modestly increased by RB1 knockdown, while there were no changes in the other apoptosis related proteins examined. Notably, NCI-H2030 cells have a G262V mutation in *TP53* that may be a basis for their high basal Survivin and minimal response to RB1 loss (see below). Further studies are clearly needed to address how effects of *RB1* loss on apoptosis related genes and on responses to BH3 mimetics are modulated by other concurrent genomic alterations.

### Decreased Survivin is dependent on p53 activation

Activation of ATR and ATM, and downstream activation of CHK1 and CHK2, can result in p53 phosphorylation and activation[40,41]. Notably, p53 stimulates the expression of PUMA and NOXA, and has been reported to suppress expression of Survivin[42]. Therefore, although we did not observe an increase in NOXA in response to nolatrexed, we assessed p53 as a potential mediator of the sensitization to navitoclax by thymidylate synthase inhibition. Nolatrexed increased p53 expression and activity, as indicated by increased expression of p21 (Fig. 6a). Moreover, the time course for increased p53 was consistent with ATR activation, as assessed by RPA32 phosphorylation (Fig. 6b). The related thymidylate synthase inhibitor raltitrexed also induced p53 activity, which was associated with increased PUMA and decreased Survivin (Fig. 6c). Importantly, this induction of p21 by nolatrexed and by raltitrexed could be prevented by thymidine supplementation (Fig. 6d). Similarly, 5-fluorouracil (5-FU), which also inhibits thymidylate synthase, synergized with navitoclax, increased p21 and decreased Survivin, with no effect on MCL-1 (Fig. 6e, f). We next used the MDM2 inhibitor nutlin to directly increase p53, and confirmed it caused a marked decrease in Survivin, with no effect on MCL-1 (Fig. 6g, left panel). Consistent with the decrease in Survivin, nutlin sensitized to apoptosis induced by navitoclax and AZD4320 (Fig. 6g, middle and right panels).

We next used siRNA to deplete p53 in LNCaP cells, which as expected abrogated the induction of p21 in response to 5-FU (Fig. 6h, left panel). Moreover, Survivin was not decreased by 5-FU in the p53 depleted cells, and was instead increased. Identical results were obtained in A549 cells (Fig. 6h, right panel). The molecular basis for p53 suppression of Survivin has not been clear. Previous studies have reported that p53 suppresses directly by binding to sites in the *BIRC5* gene, but other studies have failed to detect binding and suggested an indirect mechanism[42-45]. Notably, one

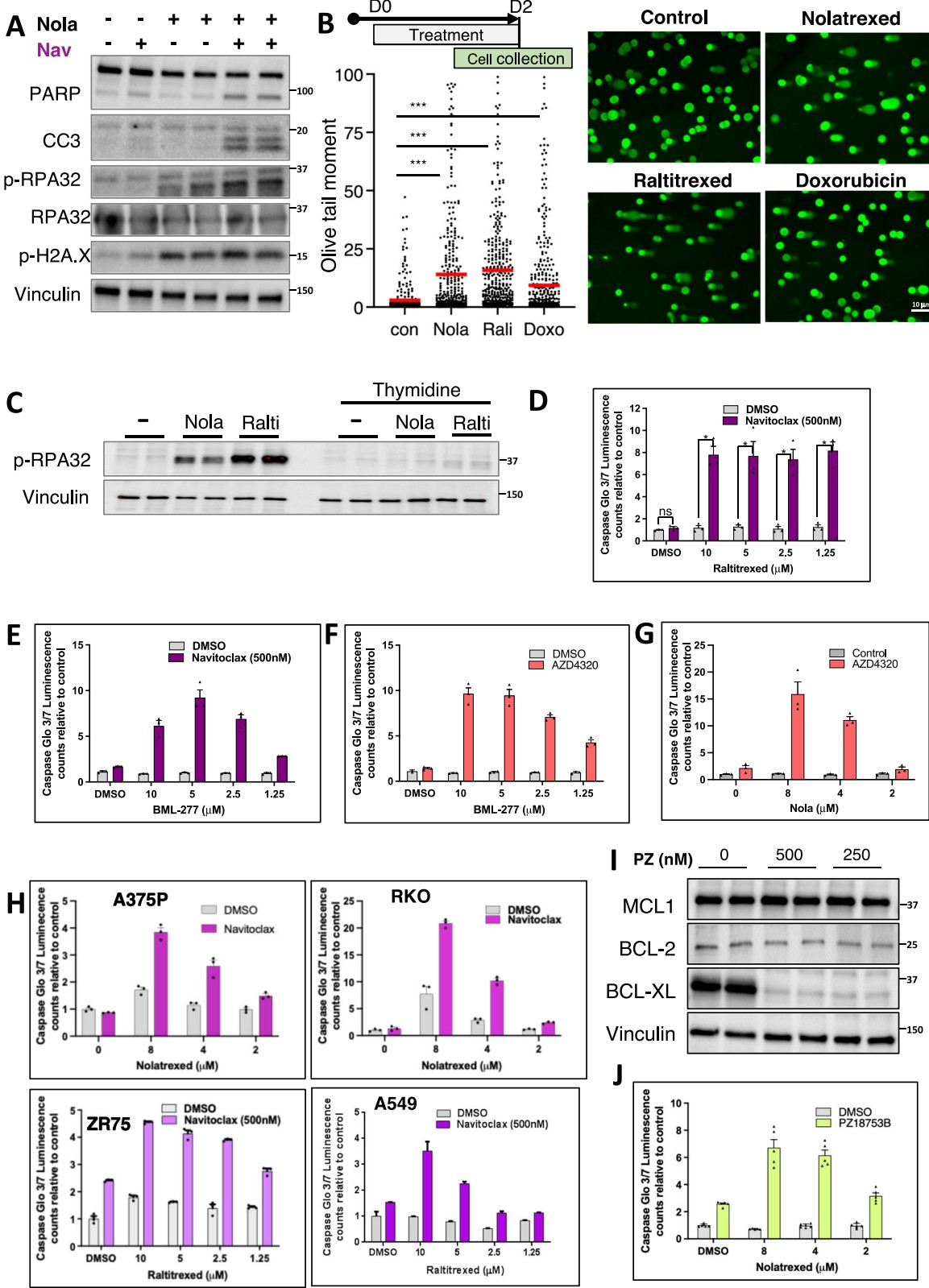

proposed indirect mechanism is through p21 dependent suppression of *BIRC5* gene expression by the DREAM complex[46]. Therefore, we next used siRNA to determine whether the suppression of Survivin by 5-FU was p21 dependent. Indeed, downregulation of p21 impaired the 5-FU mediated suppression of Survivin (Fig. 6i). These findings support an indirect mechanism through the DREAM complex.

Consistent with this p53/p21 dependent mechanism, we found that 5-FU did not sensitize to navitoclax in a series of p53 mutant cell lines (Supplementary Fig. S10a). Moreover, 5-FU did not decrease Survivin in these cells (Supplementary Fig. S10b). We also immunoblotted for pRPA32 and pH2A.X, which confirmed that there was replication stress in response to the 5-FU (Supplementary Fig. S10c). In contrast to these results, we found that nolatrexed and 5-FU sensitized

**Fig. 4 | Pharmacologic induction of replication stress sensitizes solid tumor cells to BCL-XL inhibitors. A** Immunoblotting analysis of replication stress and apoptosis markers in LNCaP cells treated with nolatrexed (48 h) followed by navitoclax for 6 h. **B** Single cell electrophoresis (Comet assay) of LNCaP cells treated for 48 h with DMSO, nolatrexed (1 μM), raltitrexed (1 μM), or doxorubicin (250 nM). Data were analyzed by one-way ANOVA ***$p < 0.001$ ($p = 0.0007$). Scale bar is 10 μm. **C** Immunoblotting analysis of cells treated with nolatrexed or raltitrexed, alone or combined with thymidine rescue for 48 h. **D** Caspase 3/7 activity in LNCaP cells treated with raltitrexed and navitoclax. Mean and SEM for 3 biological replicates are shown. Data at each raltitrexed concentration were analyzed by unpaired *t*-test, *$p < 0.05$. Two-way ANOVA showed effect of raltitrexed was significant, $p < 0.001$. **E**, **F** Caspase 3/7 activity in LNCaP cells treated with BML-277 plus navitoclax (E) or AZD4320 (F). Both were significant by two-way ANOVA, $p = 0.0001$.

**G** Caspase 3/7 activity in LNCaP cells treated with nolatrexed and AZD4320. Mean and SEM for 3 biological replicates are shown. $p < 0.0001$ by two-way ANOVA. **H** Caspase 3/7 activity in indicated cell lines (A375P - melanoma, RKO - poorly differentiated colorectal carcinoma, ZR75 - breast ductal carcinoma, A549 - lung carcinoma) treated with nolatrexed or raltitrexed for 48 h followed by navitoclax for 6 h. Mean and SEM for 3 biological replicates are shown. Two-way ANOVA showed effects of nolatrexed and raltitrexed were significant, $p < 0.001$. **I** Immunoblotting of BCL-2 family proteins in LNCaP cells treated with PZ18753B (BCL-XL/BCL2 degrader) for 24 h. **J** Caspase 3/7 activity in LNCaP cells treated with nolatrexed (48 h) and PZ18753B (16 h). Data are presented as mean values +/− SEM. Data analyzed by two-way ANOVA showed significant effect of nolatrexed, $p < 0.001$. Source data are provided as a Source Data file.

to navitoclax in p53 deficient T47D breast cancer cells (Supplementary Fig. S11a). Therefore, we carried out further mechanistic studies in these cells. While nolatrexed caused a DNA damage response in T47D cells (as assessed by increased H2A.X phosphorylation), it did not decrease expression of Survivin, indicating it was increasing BCL-XL dependence by a distinct mechanism (Supplementary Fig. S11b).

Notably, in addition to a DNA damage response, nolatrexed in the T47D cells also increased the phosphorylation of eIF2α, indicating it was activating an integrated stress response. Significantly, we reported previously that one consequence of the integrated stress response was to enhance degradation of MCL-1, and thereby sensitize to BH3 mimetics targeting BCL-XL[22]. Indeed, nolatrexed decreased MCL-1 in T47D cells (Supplementary Fig. S11b). We also assessed T47D responses to 5-FU, and found that it similarly increased eIF2α phosphorylation (Supplementary Fig. S11c), and this was associated with a decrease in MCL-1 (Supplementary Fig. S11d). In contrast, eIF2α phosphorylation or decreased MCL-1 were not observed in response to 5-FU in the other p53 deficient cells examined (see Supplementary Fig. S10b). Together these results support the conclusion that the decrease in Survivin in response to replication stress is p53 dependent, with the T47D findings showing that replication stress may in some cases increase dependence on BCL-XL by distinct mechanisms in p53 deficient cells. Notably, the BIDPC1 PDX, which was highly responsive to navitoclax, is also *TP53* deficient. Further studies are needed to determine whether there are additional mechanisms that sensitize to navitoclax in cells that are *RB1* deficient or in response to agents that cause replication stress.

### Combined inhibition of thymidylate synthase and BCL-XL is effective in vivo

To assess in vivo efficacy, we established LNCaP xenografts subcutaneously in immunodeficient male mice. When tumors reached ~500 mm³, mice were randomized to vehicle, raltitrexed, navitoclax, or the combination. The single agent treatments had no significant effect, but the combination resulted in tumor regression (Fig. 7a). Moreover, although therapy was stopped after 2 weeks, the survival of combination treated mice was greatly increased (Fig. 7b). Immunoblotting of tumor lysates confirmed that the combination therapy induced apoptosis (Fig. 7c), and analysis of the tumors in mice treated with single agent raltitrexed showed downregulation of Survivin (Fig. 7d).

Nolatrexed and raltitrexed are not generally used for cancer chemotherapy, but 5-FU or capecitabine (a prodrug for 5-FU) are commonly used to treated breast and gastrointestinal cancers. Therefore, we next assessed the in vivo efficacy of capecitabine in combination with navitoclax in ZR75 breast cancer xenografts. We first confirmed in vitro that 5-FU in ZR75 cells reduced Survivin (Fig. 7e) and sensitized to navitoclax (Fig. 7f). Xenografts were then established in nude mice, and mice were randomized to vehicle, single agent capecitabine or navitoclax, or the combination. There was no effect of single agent capecitabine, and a modest transient response to single agent navitoclax (Fig. 7g). In contrast, tumor growth was arrested by the combination. Therapy was stopped after 3 weeks and mice were

followed off therapy, which showed a modest survival advantage for single agent capecitabine and navitoclax, but markedly longer survival for the combination (Fig. 7h). Finally, immunoblotting of tumor lysates confirmed that single agent capecitabine decreased expression of Survivin (Fig. 7i).

## Discussion

BH3 mimetic drugs targeting BCL-2 alone, BCL-2 and BCL-XL, or MCL-1 have had limited efficacy in most solid tumors. This appears to reflect in part the potent antiapoptotic activity of both BCL-XL and MCL-1 in these tumors, as treatment with a BCL-XL inhibitor combined with an MCL-1 inhibitor can yield marked apoptotic responses[8,10–14]. Moreover, agents that have been reported to sensitize tumors to BCL-XL inhibition generally act by decreasing the transcription or translation of MCL-1, or by increasing its degradation[8,14,17–23]. We screened a panel of prostate cancer models and found that responses to BCL-XL inhibition were associated with *RB1* loss. In parallel, we carried out screens for drugs that would sensitize tumors to navitoclax, which identified agents that can cause replication stress, including thymidylate synthase inhibitors that act through disruption of nucleotide pools. These findings together indicated that replication stress may increase dependence on BCL-XL. We confirmed that thymidylate synthase inhibition was causing a replication stress response, but did not find alterations in the expression of BCL-2, BCL-XL, or MCL-1, or of other pro-apoptotic BH3 proteins. In contrast, there was decreased expression of Survivin, and we confirmed that decreased Survivin markedly sensitized to apoptosis in response to BCL-XL inhibition. Finally, therapy with navitoclax, in combination with the thymidylate synthase inhibitors raltitrexed or the clinically prevalent drug capecitabine, caused marked and prolonged tumor regression in prostate and breast cancer xenograft models. These findings identify a mechanism for sensitizing to BH3 mimetic drugs that may be broadly applicable for therapy of solid tumors.

While BH3 mimetic drugs are generally ineffective in solid tumors, a subset of solid tumors have genomic alterations that may increase dependence on BCL-XL or MCL-1. This includes loss of an MCL-1 ubiquitin ligase (*MARCH5*) or amplification of the *MCL-1* gene[22,47]. Previous studies have not specifically linked *RB1* loss to increased BCL-XL dependence, but have found increased sensitivity to a number of other agents including inhibitors of Aurora A and Aurora B kinases[48–50]. Notably, while the vulnerability to BCL-XL inhibition is consistent with loss of the G1/S checkpoint and subsequent replication stress, many of these vulnerabilities may reflect functions of RB1 protein that go beyond regulation of the E2F transcription factors and the G1/S checkpoint[51,52]. The conclusion that *RB1* loss increases dependence on BCL-XL is also strongly supported by large scale drug screening data from the Sanger Institute Genomics of Drug Sensitivity in Cancer and Broad DepMap. Interestingly, a recent study indicated that neuroendocrine prostate cancers, which frequently have *RB1* loss, may have increased sensitivity to BCL-2/BCL-XL inhibition, but the dependence on *RB1* status was not addressed[53].

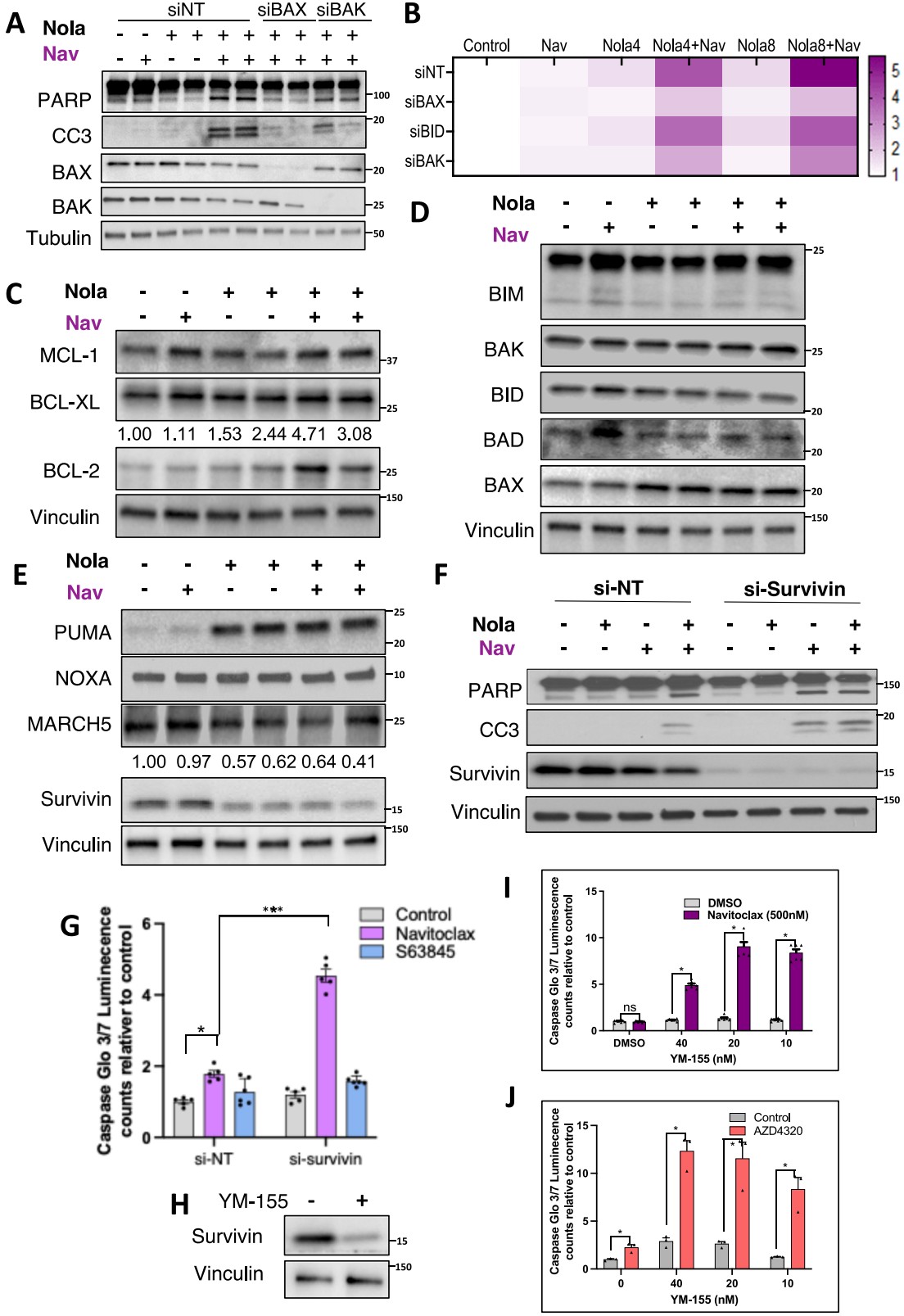

In addition to thymidylate synthase inhibitors, the drug screen identified many agents shown previously to decrease MCL-1. This included tyrosine kinase inhibitors, which at high concentrations activate an integrated stress response that increases NOXA and subsequently drives the MARCH5 dependent degradation of MCL-1[15,22]. It also included drugs that disrupt the mitotic spindle, which cause

mitotic arrest and similarly drive MCL-1 degradation through MARCH5[23]. CDK inhibition may similarly drive MCL-1 degradation via mitotic arrest (CDK1), and CDK2 inhibition has also been reported to increase MCL-1 degradation. Notably, one well-established effect of CDK9 inhibition is to decrease MCL-1, reflecting the short half-life of MCL-1 protein and mRNA[35–37]. As indicated above, we focused on

**Fig. 5 | Replication stress sensitization to navitoclax is mediated by decreased BIRC5/Survivin. A** LNCaP cells were treated with siRNA targeting BAX or BAK for 24 h followed by 48 h treatment with nolatrexed (4 μM), and addition of navitoclax for last 6 h. Whole cell lysates were then immunoblotted as indicated. **B** LNCaP cells were treated as in **A** with nolatrexed (4 or 8 μM). Apoptosis was quantified by CaspaseGlo 3/7 assay and displayed as a heat map. **C** Analysis of anti-apoptotic proteins in LNCaP cells treated with nolatrexed and navitoclax. Fold change in BCL-2 is quantified. **D** Analysis of pro-apoptotic proteins in LNCaP cells treated with nolatrexed and navitoclax. **E** Analysis of further pro- and anti-apoptotic genes in LNCaP cells treated with nolatrexed and navitoclax. Fold change in Survivin is quantified. **F** Survivin/*BIRC5* was silenced with siRNA (versus nontarget siRNA) for 3 days in LNCaP cells and apoptosis markers after nolatrexed and navitoclax treatment were assessed with immunoblotting. **G** Survivin/BIRC5 was silenced with siRNA and Caspase 3/7 activity in response to navitoclax and S63845 was assessed. Mean and SEM for 5 biological replicates are shown. Data were analyzed by unpaired *t*-test, ***$p < 0.001$. **H–J** LNCaP cells treated with YM-155 were immuno-blotted (**H**) and assessed for Caspase 3/7 activity in combination with navitoclax (**I**) or AZD4320 (**J**). Mean and SEM for 5 biological replicates are shown. Data at each YM-155 concentration were analyzed by unpaired *t*-test, *$p < .05$. Analysis by two-way ANOVA showed significant enhancement of apoptosis by YM-155 in **I** and **J**, $p < 0.001$ in both. Source data are provided as a Source Data file.

thymidylate synthase inhibitors as similarly to *RB1* they can cause replication stress, but mechanisms by which they may increase BCL-XL dependence had not been determined. Moreover, they can selectively disrupt deoxynucleotide pools, and thereby minimizes effects due to disruption of ribonucleotide pools and impairment of transcription. Using both nolatrexed and raltitrexed, we confirmed that thymidylate synthase inhibition sensitized cells to apoptosis mediated by BH3 mimetics targeting BCL-2/BCL-XL (navitoclax, AZD4320, and PZ18753B). Moreover, this was associated with increased DNA damage and a replication stress response.

We next determined that this sensitization was not associated with a decrease in MCL-1, but found increases in PUMA and a decrease in Survivin. Moreover, we confirmed that the decrease in Survivin by siRNA or pharmacologically with YM-155 could similarly sensitize to BCL-XL inhibition with navitoclax or AZD4320. The genes encoding PUMA (*BBC3*) and NOXA (*PMAIP1*) are generally induced by p53, and previous studies have shown that activation of ATR and ATM downstream of DNA damage results in phosphorylation and stabilization of p53[54]. Indeed, although we did not see an increase in NOXA, we confirmed that p53 expression and activity (based in increased p21 expression) were increased by nolatrexed. Moreover, we confirmed that treatment with nutlin-3 to block p53 degradation sensitized to navitoclax and AZD4320, and also decreased Survivin. Notably, p53 has been reported previously to suppress *BIRC5* (encoding survivin) gene expression, but the mechanism has not been clear and may be indirect[42–45]. Our results support an indirect mechanism mediated by p21 through the DREAM complex[46].

A model outlining broad mechanisms through which solid tumor cells may sensitized to BCL-XL inhibition, including through increases in BIM or decreases in MCL1 or Survivin, is shown in Supplementary Fig. S12. The decrease in Survivin in response to replication stress is shown to be p53/p21 dependent, which appears to be generally the case. However, thymidylate synthase inhibition also sensitized to navitoclax in *TP53* deficient T47D cells, and this was not associated with downregulation of Survivin. Instead, we found activation of the integrated stress response (based on phosphorylation of eIF2α) and subsequent downregulation of MCL-1. It remains to be determined why the integrated stress response is activated in the absence of p53 in these cells, but not more generally in *TP53* deficient cells.

Importantly, we confirmed that thymidylate synthase inhibition with raltitrexed in vivo in a prostate cancer xenograft could decrease Survivin and sensitize to navitoclax, with the combination causing marked and persistent tumor regression. Moreover, as nolatrexed and raltitrexed are not frequently used for cancer chemotherapy, we assessed the effects of capecitabine, which is commonly used for breast and colorectal cancer. Capecitabine, as a prodrug for 5-flur-ouracil, acts as an inhibitor of thymidylate synthase, although it and 5-flurouracil also have additional actions. Significantly, in a breast cancer xenograft model, capecitabine in combination with navitoclax caused marked and prolonged suppression of tumor growth.

Together these findings provide strong support for clinical trials of capecitabine or related agents in combination with BCL-XL inhibitors in solid tumors. Notably, while thrombocytopenia is an on-target limiting toxicity of navitoclax and other agents targeting BCL-XL, there are now newer formulations that can mitigate this toxicity[55]. Moreover, there are also alternative agents that target BCL-XL for ubiquitylation and degradation, but spare megakaryocytes due to absence of the targeted ubiquitin ligase[55]. These studies also suggest that drugs targeting Survivin, or possibly other related proteins, may be effective in combination with BCL-XL inhibition. Indeed, a very recent study found that *RB1* deficient cells had increased sensitivity to birinapant, which targets the inhibitors of apoptosis (IAP) family of proteins[56]. Finally, while we find that replication stress generally sensitizes to BCL-XL inhibition, it is possible that it will sensitize to MCL-1 inhibitors in some tumors that are more MCL-1 dependent.

## Methods

This research complies with all relevant ethical regulations of Beth Israel Deaconess Medical Center Institutional Review Board and Institutional Animal Care and Use Committee.

### PCa in vitro models

Cells were purchased from ATCC and maintained in RPMI 1640 with L-Glutamine (Corning, #MT10040CV) or DMEM with L-Glutamine and 4.5 g/L and Sodium Pyruvate (Corning, #MT10013CV), both supplemented with 10% fetal bovine serum (Gibco, #A3160401). The CaPan-1 and CaPan-1BRCA2 isogenic lines were a kind gift from the lab of Erika T. Brown, and maintained in DMEM supplemented with 20% fetal bovine serum. Cell lines were generally employed for <25 passages before fresh stocks were thawed, and cell identity was confirmed by STR profiling (ATCC) for cells passaged longer. Cells were tested monthly for Mycoplasma using the MycoAlert Kit (Lonza, ##LT07-218) according to the manufacturer's instructions.

BIDPC1-7 were generated from metastatic tumor biopsies or from rapid autopsies. Rapid autopsy was performed under IRB-approved protocol 15-441 to which patients provided written informed consent and all studies were conducted in accordance with the Declaration of Helsinki. LuCaP models have been previously described elsewhere[57,58]. Tissue fragments from prostate cancer organoids and autopsy samples underwent enzymatic digestion with Accumax (ThermoFisher). Cell clusters were transferred to Matrigel (growth factor reduced) covered tissue cultured plates and allowed to grow for 1 week. Then, Matrigel was digested with tissue recovery solution and spheroids were seeded to 96 well plate for drug screening and viability assay. Experimental drugs were added 24 h after seeding. Treatment duration varied based on experiment. Viability assays were carried out using the Cell Titer-Glo 3D assay (Promega). For primary cultures tissue fragments from organoids underwent enzymatic digestion as described above. Cell clusters were directly transferred to 96 well plate for drug screening and viability assay. Viability assays were carried out using the Cell Titer-Glo assay (Promega). For cell lines, cells were transferred to 96 well plates and grown in ATCC indicated media. Viability assays were carried out using the Cell Titer-Glo assay

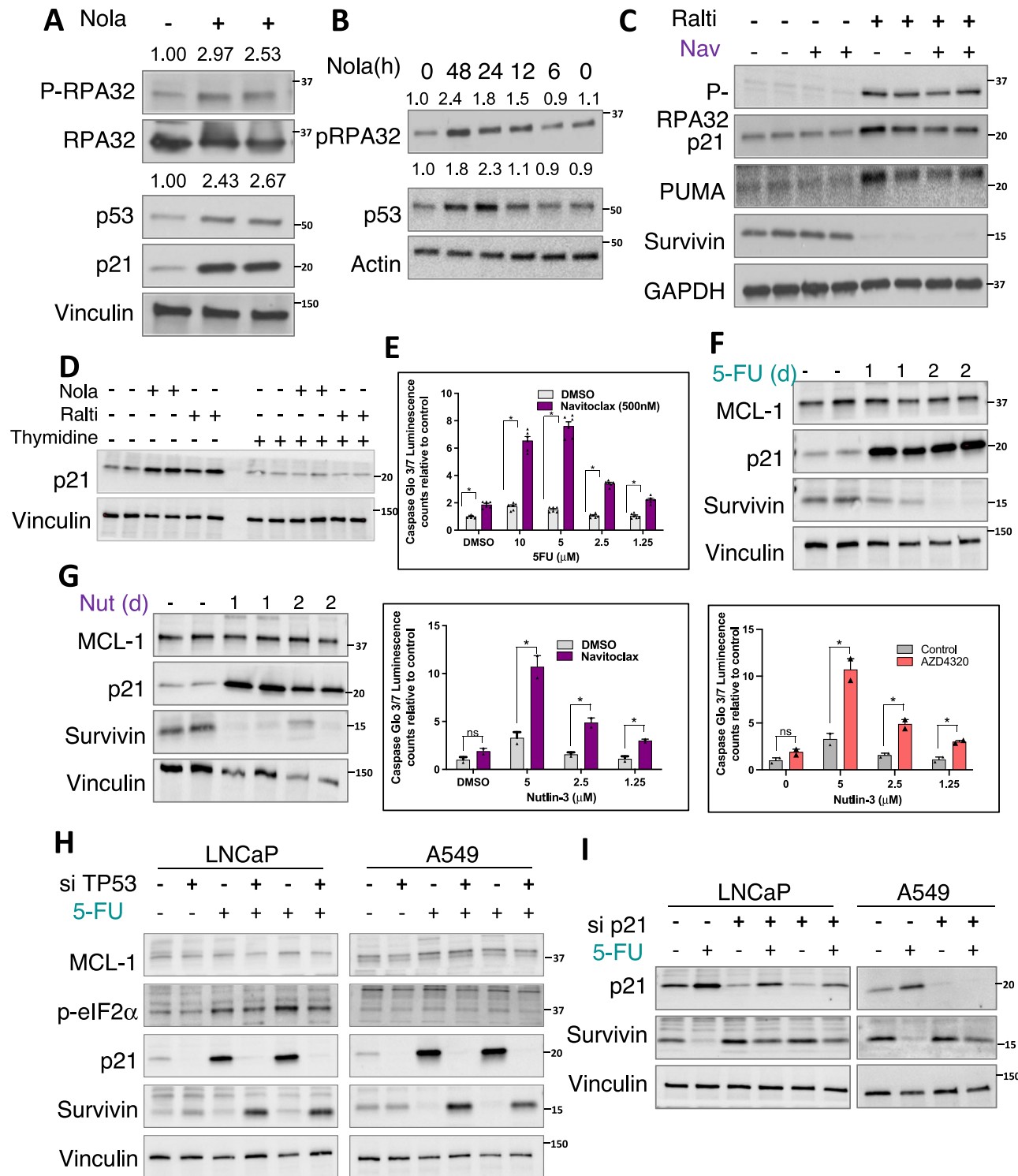

(Promega). All experiments were performed in technical replicates (at least 6 technical replicates per treatment group) and biological triplicates.

**Cell lines and RNAi**

Cells were purchased from ATCC and maintained in RPMI 1640 with L-Glutamine (Corning, #MT10040CV) or DMEM with L-Glutamine and 4.5 g/L and Sodium Pyruvate (Corning, #MT10013CV), both supplemented with 10% fetal bovine serum (Gibco, #A3160401). The CaPan-1 and CaPan-1BRCA2 isogenic lines were a kind gift from the lab of Erika T. Brown, and maintained in DMEM supplemented

with 20% fetal bovine serum. Cell lines were generally employed for <25 passages before fresh stocks were thawed, and cell identity was confirmed by STR profiling (ATCC) for cells passaged longer. All cell lines reported in this manuscript tested negative for mycoplasma contamination. The siRNA targeting BIRC5 were from Cell Signaling Technologies (SignalSilence Survivin siRNA II #6546). All others were from Horizon Discovery Biosciences Ltd/Dharmacon. These included siRNA targeting RB1 (L-003296-02-0005), BAK (L-003305-00-0005), BAX (L-003308-01-0005), PUMA (L-004380-00-0005), TP53 (E-003329-00-0005), and CDKN1A (L003471-00-0005).

**Fig. 6 | Activation of p53 mediates survivin depletion and sensitivity to BCL-XL inhibitors. A** LNCAP cells treated with nolatrexed (2 µM) for 48 h were immunoblotting for DNA damage response and p53 activation. Fold changes in P-RPA32 and p53 are quantified. **B** Time-course for DNA damage response and p53 induction in nolatrexed treated LNCaP cells. Fold changes in P-RPA32 and p53 are quantified. **C** Immunoblotting for DNA damage response and p53 targets in cells treated with raltitrexed and/or navitoclax. **D** Immunoblotting for p21 in LNCaP cells treated with thymidylate synthase inhibitors with and without thymidine rescue. **E** LNCaP cells were treated with 5-FU for 2 days followed by navitoclax for 6 h and caspase activation was assessed. Mean and SEM for 6 biological replicates are shown. Data at each 5-FU concentration were analyzed by unpaired $t$-test, $*p < .05$. Analysis by two-way ANOVA showed significant effect of 5-FU, $p < 0.001$. **F** Immunoblotting for MCL-1, p21, and survivin in LNCaP cells treated with 5-FU for 1–2 days. **G** LNCaP cells were treated with Nutlin-3a (MDM2 inhibitor) for 1–2 days. Left panel: LNCaP cells were treated with Nutlin-3a for 1–2 days followed by immunoblotting. Middle and left panels: LNCaP cells were treated for 24 h with Nutlin-3a followed by 6 h with navitoclax (middle panel) or AZ4320 (right panel) and assessed for apoptosis by Caspase Glo 3/7 assay. Mean and SEM for 3 biological replicates are shown. Data at each Nutlin-3 concentration were analyzed by unpaired $t$-test, $*p < .05$. Analysis by two-way ANOVA showed effects were significant ($p = 0.006$ for left panel and $p = 0.0013$ for right panel). **H** LNCaP and A549 cells were treated with TP53 or nontarget control siRNA. The effect of 5-FU on the p53 pathway and integrated stress response pathway was assessed with immunoblotting. **I** LNCaP cells were treated with p21 or nontarget control siRNA. The effect of 5-FU on survivin expression was then assessed with immunoblotting. Source data are provided as a Source Data file.

## Immunoblotting

Cells were lysed in RIPA buffer (#PI89900, Fisher Scientific) supplemented with protease inhibitor (#PI78437, Fisher Scientific) and phosphatase inhibitor cocktails (#PI78426, Fisher Scientific). The following primary antibodies were used for immunoblotting at 1:1000 dilution unless otherwise indicated: anti-RPA32/RPA2 (phospho S33, Abcam #ab211877, 1:2000), anti-H2A.X phospho Ser139 (clone 20E3, Cell Signaling Technology #9718), anti-p53 (Cell Signaling Technology #9282), anti-p21 (clone EPR3993, Abcam #ab109199), anti-Survivin (Cell Signaling Technology #2808), anti-RB1 (clone 4H1, Cell Signaling Technology #9309, 1:2000), anti-Bad (#9239, Cell Signaling Technology, 1:500), anti-BAK (#12105, Cell Signaling Technology), anti-BAX (#5023, Cell Signaling Technology), anti-β-actin (#ab6276, Abcam), anti-BCL2 (#4223, Cell Signaling Technology, 1:500), anti-BCL-XL (#2764, Cell Signaling Technology), anti-BIM (#2933, Cell Signaling Technology), anti-cleaved caspase 3 (#9664, Cell Signaling Technology1:250), anti-MARCH5 (#06-1036, EMD Millipore, 1:500), anti-MCL1 (#5453, Cell Signaling Technology), anti-NOXA (#ab13654, Abcam, 1:250), anti-PARP (#9532, Cell Signaling Technology), anti-phospho-eIF2α Ser51 (#9721, Cell Signaling Technology), anti-PUMA (#12450, Cell Signaling Technology, 1:500), or anti-vinculin (#sc-73614, Santa Cruz Biotechnology, 1:20,000). The secondary antibodies were 1:5000 of anti-rabbit (#W401B) or anti-mouse (#W402B) secondary (Promega).

## Immunohistochemistry

Immunohistochemistry for RB1 was performed on various patient derived xenograft (PDX) passages [CP50C ($n = 3$), CP253C ($n = 2$), CP267C ($n = 1$), CP336C ($n = 3$)] using the mouse anti-RB1 monoclonal antibody (clone 4H1, Cell Signaling Technology, Massachusetts, USA). Normal prostate tissue was used as a positive control. Cell pellets from 22Rv1 cells treated with control or Rb1 siRNA were used to confirm specificity of the antibody. Mouse IgGs were used as negative controls. Nuclear quantification for each sample was determined by a pathologist (author B.G) blinded to clinical and molecular data using modified H-score ([% of negative staining x 0] + [% of weak staining x 1] + [% of moderate staining x 2] + [%of strong staining x 3]), to determine the overall percentage of positivity across the entire stained sample, yielding a range from 0 to 300.

## Mechanism of Action library drug screening

Drug screens were carried out in the ICCB-Longwood Screening Facility using their Mechanism of Action Library (ICCB-L MoA), which contains 1245 compounds. LNCAP cells (confluency 2000 cells per well) were seeded in 384-well plates in 30 µl of RPMI media (10% FBS, antibiotic free) containing DMSO or navitoclax (500 nM). Cells were allowed to grow for 2 days. Mechanisms of action library compounds were then added to plates at 4 concentrations in duplicate. Cell growth was analyzed after another 2 days in culture using Cell Titer-Glo viability assay (Promega). Active compounds from the primary screen were selected based on z-score ($<-1.5$) comparing rank of single agents compared to the combination with navitoclax. Further DOI selection was performed based on biological relevancy of target.

## Comet assay (Single-cell electrophoresis)

Alkaline single-cell electrophoresis was performed using Comet Assay High Throughput kit (Trevigen, R&D 4252-040-K) following the manufacturer's instruction. Briefly, prostate cancer cell lines DU-145 and LNCaP were treated as indicated, trypsinized, pellets were collected and washed twice in cold PBS. Later, cells were diluted in PBS w/o Ca$^{2+}$ and Mg$^{2+}$ and diluted 1:10 with low melting agarose. A drop of agarose/cell mix was spread on a comet assay slide, solidified at 4 °C in the dark, and submerged into lysis buffer overnight at 4 °C. The next day comet slide was incubated in alkaline unwinding solution for 20 min at RT and subjected to electrophoresis at 17 V for 25 min for DU145 cells and at 17 V for 34 min for LNCaP cells. After the run cells were washed in distilled water and 70% ethanol, dried at 37 °C for 6 h, and stained with SyBr Gold nucleic acid stain (Thermo Fisher S11494). Images were taken on Nikon Eclipse TE2000S fluorescent microscope using a 10x objective. Comets were evaluated using CometScore 2.0 free software and the Olive Tail Moment value[59] of each individual cell was plotted using Graph Pad Prism Software v8, with red lines portraying the mean OTM value of each treatment. Mann–Whitney test was used for the evaluation of statistical significance.

## DNA fiber spreading

LNCaP cells were treated with nolatrexed (2 µM or 4 µM) or DMSO for 16 h in 10% FBS RPMI. DNA fibers were performed as described previously[60]. Briefly, cells were sequentially pulsed with two thymidine analogs, 50 µM CIdU (Sigma, C6891) and 150 µM IdU (Sigma, 17125), with 2xPBS washes in between. Cells were then trypsinized and resuspended in PBS, and 2.5 µL were pipetted on the top of SuperFrost plus slides (#48311-703, VWR). After 4 min, 7.5 µL spreading buffer (0.5% SDS, 200 mM Tris-HCl pH 7.4, 0.5 mM EDTA) was mixed with the cells for an additional 2 min. Two glass slides were made per condition per experiment. Slides were tilted at 15 degrees to allow DNA fibers to run down the glass slide. Later, the fibers were air dried and then fixed in 3:1 methanol:acetic acid solution for 2 min, followed by the 2.5 M HCl treatment for 30 min and 3% BSA/PBST blocking for 1 h. Primary antibody incubation was performed for 1 h with anti-CIdU (ab6326 Abcam, 1:100) and anti-IdU (BD-347580, 1:20). Following three washes with PBS, fibers were stained with appropriate secondary Alexa-Fluor conjugated antibodies for 30 min, washed, air-dried and mounted. Slides were imaged with Zeiss LSM 880 Upright Confocal System, 63x PlanApo oil immersion objective. Measurement of replication structures was performed using Fiji. At least 200 fiber tracks were quantified per experimental condition per assay. Replication fork speed was calculated as described in by using the conversion factor 1 µm = 2.59 kb[61].

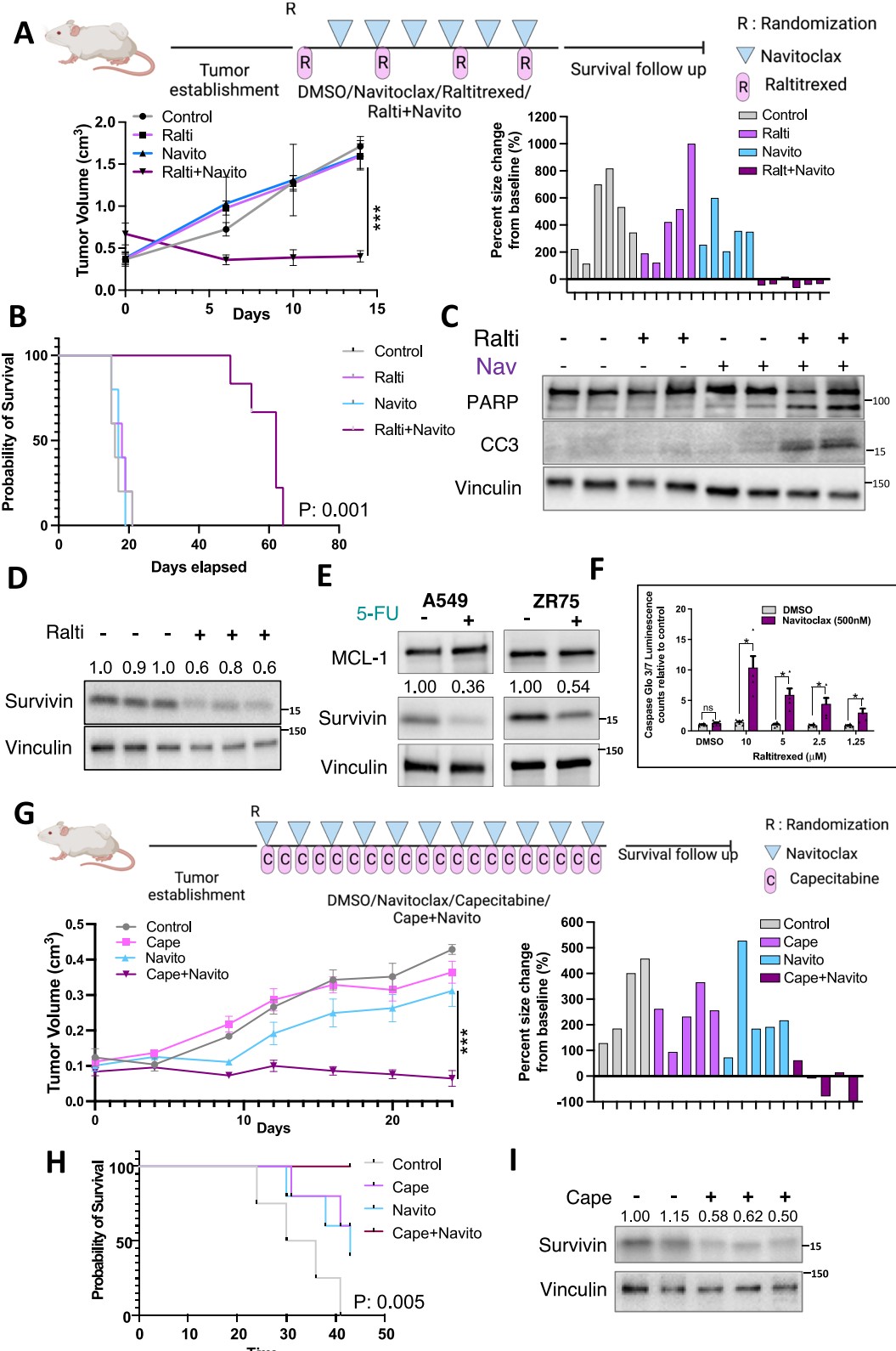

## Animal models

For all experiments we used 6–8 week old ICRSC-M, IcrTac:ICR-*Prkdc^scid* mice (Taconic). Light cycle was a 14 h light/10 h dark cycle. Temperature was 65–75 °F (-18-23 °C) with 40–60% humidity. In vivo experiments were performed under the BIDMC IACUC approved protocol-051-2022. The PDXs (BIDPC1 and BIDPC5) were developed from patient tumor fragments obtained during removal of a vertebral mass (BIDPC1) or from liver in a rapid autopsy (BIDPC5). Rapid autopsies were performed under BIDMC IRB-approved protocol 15-441 and all studies were conducted in accordance with the Declaration of Helsinki.

The BIDPC5 PDX has been described previously, and in addition to *RB1/BRCA2* loss has an activating *EGFR* mutation[62]. The BIDPC1 PDX has biallelic chromosomal loss of 13q13.1 – 13q14.2 encompassing *BRCA2*

**Fig. 7 | Combined inhibition of thymidylate synthase and BCL-XL is effective in vivo. A** LNCaP xenografts were developed in immunodeficient male mice. Mice were randomized to control, raltitrexed (50 mg/kg q4 days), navitoclax (50 mg/kg q2 days), or the combination as indicated in schema. Actual tumor size measurements are shown on left. Individual tumor size changes at 14 days relative to baseline (tumor size at randomization) are shown on right panel. Five (5) biological replicates are shown. Data were analyzed by Wilcoxon Rank Sum Test ***$p < 0.001$. Upper panel was created in Biorender. Yuan, X. (2025) https://BioRender.com/n41n675 **B** Tumor-bearing mice were treated as in (A) for 14 days were monitored for tumor growth and euthanized when tumors reached 2.0 cm$^3$. Graph shows Kaplan-Meier curve for mice survival from randomization. Data were analyzed by logrank test. **C** Tumors in another cohort were collected 6 h after the second raltitrexed dose (combination treated mice received 2 navitoclax doses). Apoptosis markers (cleaved PARP and cleaved caspase 3, CC3) were assessed with immunoblotting. **D** Tumors were collected 6 h after second raltitrexed dose and survivin protein expression was assessed with immunoblotting. Fold change in survivin is quantified. **E** Immunoblotting for MCL-1 and survivin in A549 (lung adenocarcinoma) and ZR75 (breast cancer) cells treated with 5-FU. Fold change in survivin is

quantified. **F** ZR75 cells were treated with 5-FU for 48 h followed by navitoclax for 6 h, and apoptosis was assessed by CaspaseGlo 3/7 assay. Mean and SEM for 3 biological replicates are shown. Data at each 5-FU concentration were analyzed by unpaired $t$-test, *$p < .05$. Analysis by two-way ANOVA showed raltitrexed significantly enhanced apoptosis, $p < .001$. **G** ZR75 xenografts were developed in immunodeficient mice. Mice were randomized to control, capecitabine (200 mg/kg daily), navitoclax (50 mg/kg q2 days), or combination therapy as indicated in schema. Actual tumor size measurements are shown in the left panel. Individual tumor size changes at 28 days relative to baseline are shown in right panel. Five biological replicates are shown. Data were analyzed by Wilcoxon Rank Sum Test, ***$p < 0.001$. Upper panel was created in Biorender. Yuan, X. (2025) https://BioRender.com/obtzw73 **H** Tumor-bearing mice were treated for 28 days. After that, mice were monitored for tumor growth as described above. Graph shows Kaplan-Meier curve for mice survival from randomization. Data were analyzed by logrank test. **I** Tumors were collected 6 h after the fourth capecitabine dose, and survivin protein expression was assessed with immunoblotting. Fold change in survivin is quantified. Source data are provided as a Source Data file.

and *RB1*. Additional alterations identified by whole exome sequencing are 2 copy losses in *TP53*, *CHD1*, and *APC*, and a *FOXA1* frame shift (Q260fs). After model establishment, tumor fragments were preserved frozen in preservation media. For these experiments, 2$^{nd}$ generation xenografts were developed in immunodeficient mice (scid) and treated with navitoclax (50 mg/kg q2 days) for 14 days. After that, mice were monitored for tumor growth and toxicity.

For LNCaP we established LNCaP xenografts subcutaneously in immunodeficient male nude mice. When tumors reached ~500 mm$^3$, mice were randomized to vehicle (DMSO), raltitrexed (50 mg/kg q4 days), navitoclax (50 mg/kg q2 days), or the combination. Treatment was given intraperitoneally for 14 days. After that mice were monitored for tumor size and toxicity. For ZR75 we established ZR75 xenografts subcutaneously in immunodeficient mice. When tumors reached 100 mm$^3$, mice were randomized to vehicle, capecitabine (200 mg/kg daily), navitoclax (50 mg/kg q2 days) or combination. Treatment was given through oral gavage for 24 days. After that, mice were monitored for tumor size and toxicity. Treated mice were also weighed twice per week and no loss beyond 10% of baseline was noted. For analysis of apoptosis markers in tumors treated with raltitrexed (given q4 days) and capecitabine (given daily) respectively, we sacrificed mice after 4 days of treatment, and within 2 h of last dose. Protein lysates were created from frozen tumor fragments using mechanical dissociation. Apoptosis markers were analyzed by immunoblotting. The maximal tumor size permitted by our IACUC board is 2000 mm$^3$. We confirm that the maximal tumor size was not exceeded in any of the experiments described in this manuscript.

## Statistics and reproducibility

For in vitro experiments, comparison between groups was performed with Wilcoxon's rank test. For in vivo experiments, difference between groups was assessed with Wilcoxon's rank sum test. Survival analysis was performed with Kaplan–Meier product-limit method to estimate the distribution of each group and the log rank test to compare the four groups. Each individual experiment presented in this manuscript was performed at least 3 times independently with similar results.

## Reporting summary

Further information on research design is available in the Nature Portfolio Reporting Summary linked to this article.

## Data availability

Any data not included in the manuscript will be made available upon request. Materials generated in this study will be made available for academic use with an MTA. Source data are provided as a Source Data file. Source data are provided with this paper.

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

## Acknowledgements

A.V. had support from DoD (Physician Research Award, PC200820, GRANT13266620) and ASCO (Young Investigator Award, 2021A010981). S.P.B. had support from NIH (R01 CA262536 and PO1CA163227), a Koch Institute-Dana Farber/Harvard Cancer Center Bridge Project Award (S.P.B., M.G.V.H., D.R.S.), and a Prostate Cancer Foundation Challenge award (S.P.B., E.C., A.S., J.S.D.). E.C. also had support from the Pacific Northwest Prostate Cancer SPORE (P50CA97186) and the PO1 NIH grant (PO1CA163227). We would like to thank the patients who generously donated tissue that made this research possible. We thank Dr. Karen Knudsen for providing LNCaP cells expressing RB1 shRNA. We thank the staff at the ICCB-Longwood Screening Facility for assistance in the drug screening. We also thank Jennifer Conner, Conner Sessions and the Comparative Medicine Animal Caregivers for assistance with the LuCaP xenograft work.

## Author contributions

Conceptualization: A.V., D.R.S., J.S.D., M.G.V., E.C., A.S., S.P.B.; Data curation: A.V., A.S., S.P.B.; Formal Analysis: A.V., D.R.S., L.P., T.M., M.G.V., E.C., A.S., S.P.B.; Funding acquisition: A.V., J.S.D., M.G.V., E.C., S.P.B.; Investigation: A.V., K.W., A.S., D.R.S., S.A., N.A., N.K., M.N., J.M.V., D.W., J.C., F.X., I.F., L.B., A.N., B.G., N.C., L.B., O.V., S.Y.C., J.W.R., X.Y., D.J.; Methodology: T.M., M.G.V., E.C.; Project administration: S.P.B.; Resources: S.A., H.B., J.S.D., D.J.E., E.C.; Supervision: A.V., T.M., J.S.D., M.G.V., E.C., A.S., S.P.B.; Validation: A.V., S.P.B. Visualization: A.V., S.P.B.; Writing original draft: S.P.B.; Writing edits: A.V., J.S.D., E.C., A.S.

## Competing interests

The authors declare no competing interests.
