## [Transparent Peer Review file · Nature Communications]

BH3 mimetics targeting BCL-XL have efficacy in solid tumors with RB1 loss and replication stress

Corresponding Author: Dr Steven Balk

Version 0:

Reviewer comments:

Reviewer #1

(Remarks to the Author)

The manuscript by Varkaris and colleagues explores the use of BH3 mimetics, in particular BCL-xL inhibitors, in combination with thymidylate synthase inhibitors in prostate and breast cancer (mostly), which could help enhance their treatment. Overall, the study is well-written and elegantly performed. However, some aspects could be improved.

Major comments:

- The authors predominantly use navitoclax and AZD4320 as BCL-xL inhibitors, although these molecules also inhibit BCL-2 (and BCL-W in the case of navitoclax). The authors included WEHI in Fig 2, but the manuscript would benefit if specific BCL-xL inhibitors such as A-1331852, A-1155463 or similar were used to support the specific role of this antiapoptotic protein.
- How does RB1 loss affect BCL-2 family members expression, particularly antiapoptotic proteins?
- The authors focus mostly on the cell line LnCap. Is the BCL-xL dependence observed in other navitoclax resistant cell lines?
- Mice weight should be monitored to discard potential undesired toxicities from the treatments tested.

Minor comments

- In page 13, the authors mention Fig. 8D instead of 7D.
- The authors should quantify the optical density in the Western Blot analyses.

Reviewer #2

(Remarks to the Author)

Varkaris et al propose the use of BH3 mimetics in solid tumors with RB1 loss and replication stress. In general, the paper is well written but the main concern this paper raises is that replication stress is not experimentally shown. The prototypical experiment to detect DNA replication stress in vitro is the DNA fiber method. By utilizing this technique, replication tracts and fork speed can be detected and quantified. gH2AX and pRPA32 expression have been used in the paper to confirm the presence of replication stress, however those are markers of DNA damage and their expression does not imply replication stress.

The authors claim that thymidylate synthase inhibition by nolatrexed causes replication stress response, but this is never confirmed. Experiments show DNA damage. Replication stress was inferred but not shown.

In addition, TS levels were not addressed in any of the nolatrexed or raltitrexed treatments performed throughout the paper.

Neither TS catalytic activity was done to confirm lack of activity of TS and disruption of dNTP when cells were treated with nolatrexed.

Reviewer #3

(Remarks to the Author)

In this manuscript the authors provide an analysis of novel genetic determinants of sensitivity to Bcl-XI inhibitors as well as comprehensive pharmacological determinants of synergy in otherwise unresponsive tumors. The manuscript is clearly written and easy to follow. Moreover, the conclusions of the study bear novel and interesting biological insights into Bcl-XI inhibition that may potentially be broadly applicable for therapy of solid tumors (ie, uncovering replication stress as a critical determinant) and may also have important translational value in the design of novel clinical trials based on novel drug combinations, potentially in selected patients. Most experiments are well conducted and documented, although some concerns related to insufficient information and data presentation as well as statistical reporting and mentioned below.

The manuscript is comprised of two parts, one uncovering a role for Rb1 loss as determinant of sensitivity to Bcl-XI inhibition and another one identifying synergy of navitoclax with other drugs. The latter part is mechanistically convincing and thorough, using through multiple drugs and models to state that replication stress induced by nucleotide pool depletion sensitizes to Bcl-XI inhibitors through defined p53-mediated mechanisms and yet un-defined p53-independent mechanisms. The former part, however, lacks the same thorough mechanistic dissection and does not necessarily support the plausible yet unproven link between Rb1 loss, replication stress an inhibitor sensitivity.

MAJOR CONCERNS:

- The authors use a large collection of models to query the association of Navitoclax sensitivity to genetic alterations, and appear to develop a series of new model organoids (BIDP-) from metastatic sites, which is of great value. However, no clear explanation is given on the histoclinical characteristics of the patients from where these models were derived, their tissue of origin and the culture conditions. Perhaps more importantly, there is no description of the methodology used to characterize the molecular alterations, limiting the interpretability of their results.
- The connection between Rb1 loss and sensitivity to navitoclax is promising and provocative yet is not thoroughly addressed:
 - o Rb1 loss should be confirmed or analyzed by western blot/IHC to in sensitive and resistant models.
 - o Similarly, in Figure 2C,D, no explanation is given on how “Rb1alt” was defined, on how the Genomics of Drug sensitivity database was analyzed. Critically, the plotted results labeled as “Rb1alt” seem to derived from mutational and not copy number analysis of the database, which is misleading. Indeed, there seems to be no significant association when RB1 copy number is used instead (cnaPANCAN389). These results should thus be re-interpreted and more experimental validation of the association to Rb1 loss should be provided, perhaps with other databases.
- The functional link to Rb1 should also be experimentally reinforced. A single siRNA is used in a single cell line to substantiate this claim. More rigorous analyses using multiple siRNAs or rescue experiments and other cell line models should also be included.
- Is a WT p53 status required for Rb1 loss-associated sensitivity to Navitoclax?
- A major concern is that Rb1 loss is not mechanistically linked to replication stress and thus sensitize to Bcl-XI inhibition – the authors reference papers for which similar references could be found for other genetic alterations impacting genomic instability, other than Rb1. Indeed, loss of Tp53 (cnaPANCAN307) is significantly associated with sensitivity to navitoclax in the Genomics of Drug sensitivity database. The authors should study whether the same or different mechanisms of action (replication stress, stress response, etc) occur with Rb1 loss.
 - o How does Rb1 loss impact BH3 protein expression (ie, Bcl2, Bcl-XI, MCL1, etc)
 - o Does Rb1 loss lead to increased gH2aX/RPA32?
- The drug sensitivity screen is inadequately presented or explained. More detailed information should be provided for what the ICCB-L MoA library is, what drugs are included, results for different drug concentrations and, critically, how the statistical significance of the “hits” was assessed. If synergy was a criteria, how was this assessed? This is also important to assess the relative relevance of the different classes of drugs for overall sensitivity and to understand where navitoclax lies.
- The authors show in vivo data to validate the efficacy of novel drug combinations. However, the statistical presentation of the results are poor. Please show p values, n numbers and statistical tests used. Confirmation of induction of apoptosis in tumors (ie, by IHC stain) is warranted, at least at initial time points. Moreover, no toxicity studies are reported. The authors should at least document whether behavioral, body weight and hematological toxicities were observed.

MINOR CONCERN

- Some results are not presented or explained clearly. Most CTG and CaspaseGlo assays are missing statistical analyses for significance. What sensitivity parameter is plotted on the heatmap of Figure 1B? Statistical significance is not assessed.

Version 1:

Reviewer comments:

Reviewer #1

(Remarks to the Author)

The authors answered all my suggestions.

Reviewer #2

(Remarks to the Author)

The authors have adequately addressed our comments in the revised version of the manuscript and they updated the figures accordingly.

Reviewer #3

(Remarks to the Author)

I commend the authors for a thorough revision. All of my concerns have been addressed.

Reviewer #1 - BH3 mimetics (Remarks to the Author):

The manuscript by Varkaris and colleagues explores the use of BH3 mimetics, in particular BCL-xL inhibitors, in combination with thymidylate synthase inhibitors in prostate and breast cancer (mostly), which could help enhance their treatment. Overall, the study is well-written and elegantly performed. However, some aspects could be improved.

Major comments:

- The authors predominantly use navitoclax and AZD4320 as BCL-xL inhibitors, although these molecules also inhibit BCL-2 (and BCL-W in the case of navitoclax). The authors included WEHI in Fig 2, but the manuscript would benefit if specific BCL-xL inhibitors such as A-1331852, A-1155463 or similar were used to support the specific role of this antiapoptotic protein.

Response: In addition to navitoclax and AZD4320, we had shown that the PROTAC PZ18753B could drive apoptotic responses in response to nolatrexed (Figure 4I, J). While PZ18753B can target BCL2 as well as BCLXL, the extent to which it targets BCL2 is variable, and we showed that it was dramatically reducing BCLXL with no clear effect on BCL2 (Figure 4I). Nonetheless, as suggested by the reviewer, we also assessed effects of a BCLXL specific degrader (DT2216). As shown in the new figure (Figure S7B-D), DT2216 selectively depleted BCL-XL and sensitized to raltitrexed and nolatrexed.

- How does RB1 loss affect BCL-2 family members expression, particularly antiapoptotic proteins?

Response: This was not addressed in the previous manuscript. We now include immunoblotting for MCL1, BCLXL, BCL2, BAX, BIM, and survivin in MCF7 (breast cancer) and NCI-2030 (non small cell lung cancer) cell lines pre- and post RB1 knockdown (Figure S9). Interestingly, RB1 knockdown in MCF7 cells resulted in increased levels of MCL1, BIM, and BAX, while survivin (which was expressed at low basal levels) was decreased. In contrast, survivin was expressed at high basal levels in NCI-H2030 cells and was modestly increased by RB1 knockdown, while there were no changes in the other apoptosis related proteins examined. Notably, NCI-H2030 cells have a G262V mutation in TP53 that may be a basis for their high basal survivin and minimal response to RB1 loss. The extent to which these alterations in apoptosis related proteins affect apoptotic threshold is unclear, but we do also find that the RB1 knockdown MCF7 cells does not sensitize them to navitoclax, which could reflect their increase in MCL1. Based on these results, we indicate that further studies are needed to address how effects of RB1 loss on apoptosis related genes and on responses to BH3 mimetics are modulated by other concurrent genomic alterations.

- The authors focus mostly on the cell line LNCaP. Is the BCL-xL dependence observed in other navitoclax resistant cell lines?

Response: The reviewer is correct that we focused on LNCaP for mechanistic studies. However, Figure 4H in the manuscript shows that navitoclax drives apoptosis in response to thymidylate synthase inhibition in several other cell lines (A375P, RKO, ZR75, A549).

- Mice weight should be monitored to discard potential undesired toxicities from the treatments tested.

Response: We have assessed weights twice per week under each treatment regimen and did not observe any losses beyond 10% of baseline. This has now been added to the text.

Minor comments

- In page 13, the authors mention Fig. 8D instead of 7D.

Response: Thank you for picking this up. It has been corrected.

- The authors should quantify the optical density in the Western Blot analyses.

Response: We have added quantification to a subset of the blots where there is substantial expression pre- and post-treatment in order to confirm and quantify the indicated alterations in protein expression.

Reviewer #2 - Targeting thymidylate synthase (Remarks to the Author):

Varkaris et al propose the use of BH3 mimetics in solid tumors with RB1 loss and replication stress. In general, the paper is well written but the main concern this paper raises is that replication stress is not experimentally shown. The prototypical experiment to detect DNA replication stress in vitro is the DNA fiber method. By utilizing this technique, replication tracts and fork speed can be detected and quantified. γ H2AX and pRPA32 expression have been used in the paper to confirm the presence of replication stress, however those are markers of DNA damage and their expression does not imply replication stress.

The authors claim that thymidylate synthase inhibition by nolatrexed causes replication stress response, but this is never confirmed. Experiments show DNA damage. Replication stress was inferred but not shown.

Response: The reviewer is correct that we did not directly show replication stress in response to thymidylate synthase inhibition. Notably, previous studies have established that nucleotide pool disruption causes replication stress, which can then lead to a DNA damage response. Moreover, hydroxyurea, which inhibits thymidylate synthase, is frequently used as a positive control for DNA fiber assays. However, hydroxyurea is not specific for thymidylate synthase, and we agree that confirming replication stress under the conditions we are using would strengthen our conclusions. Therefore, we have carried out DNA fiber assays, and confirm in a new figure that thymidylate synthesis is causing marked replication stress (Figure S5).

In addition, TS levels were not addressed in any of the nolatrexed or raltitrexed treatments performed throughout the paper. Neither TS catalytic activity was done to confirm lack of activity of TS and disruption of dNTP when cells were treated with nolatrexed.

Response: The reviewer is correct that we did not directly measure TS levels or catalytic activity. However, we showed by mass spectrometry in Figure S4B (Figure S6 in revised manuscript) that raltitrexed was causing a dramatic decrease in its product (dTTP) and increase in its substrate (dUMP). We believe this confirms that we are obtaining robust TS inhibition.

Reviewer #3 - Prostate cancer therapy, organoids (Remarks to the Author):

In this manuscript the authors provide an analysis of novel genetic determinants of sensitivity to Bcl-XI inhibitors as well as comprehensive pharmacological determinants of synergy in otherwise unresponsive tumors. The manuscript is clearly written and easy to follow. Moreover, the conclusions of the study bear novel and interesting biological insights into Bcl-XI inhibition that may potentially be broadly applicable for therapy of solid tumors (ie, uncovering replication stress as a critical determinant) and may also have important translational value in the design of novel clinical trials based on novel drug combinations, potentially in selected patients. Most experiments are well conducted and documented, although some concerns related to insufficient information and data presentation as well as statistical reporting and mentioned below.

The manuscript is comprised of two parts, one uncovering a role for Rb1 loss as determinant of sensitivity to Bcl-XI inhibition and another one identifying synergy of navitoclax with other drugs. The latter part is mechanistically convincing and thorough, using through multiple drugs and models to state that replication stress induced by nucleotide pool depletion sensitizes to Bcl-XI inhibitors through defined p53-mediated mechanisms and yet un-defined p53-independent mechanisms. The former part, however, lacks the same thorough mechanistic dissection and does not necessarily support the plausible yet unproven link between Rb1 loss, replication stress an inhibitor sensitivity.

MAJOR CONCERNS:

- The authors use a large collection of models to query the association of Navitoclax sensitivity to genetic alterations, and appear to develop a series of new model organoids (BIDP-) from metastatic sites, which is of

great value. However, no clear explanation is given on the histoclinical characteristics of the patients from where these models were derived, their tissue of origin and the culture conditions. Perhaps more importantly, there is no description of the methodology used to characterize the molecular alterations, limiting the interpretability of their results.

Response: We agree that more information should have been provided on these PDXs and derived organoids, particularly the RB1 deficient BIDPC1 and BIDPC5 models that were examined in vitro and in vivo. As noted in the Methods section, all were derived from metastatic sites, and all were assessed by whole exome sequencing. We now add that BIDPC1 was from a vertebral metastasis and BIDPC5 from a liver metastasis. We have recently published on BIDPC5 and that reference is now included (Einstein DJ, Arai S, Calagua C, Xie F, Voznesensky O, Capaldo BJ, et al. Metastatic Castration-Resistant Prostate Cancer Remains Dependent on Oncogenic Drivers Found in Primary Tumors. JCO Precis Oncol. 2021;5). Interestingly, as described in this reference, in addition to biallelic RB1 loss in BIDPC5 (which was not found in the corresponding primary tumor), this tumor has a truncal activating mutation in EGFR. For BIDPC1, whole exome sequencing shows a biallelic chromosomal loss of 13q13.1 – 13q14.2 encompassing BRCA2 and RB1. Additional alterations are 2 copy losses in TP53, CHD1, and APC, and a FOXA1 frame shift (Q260fs). This information is now added to the Methods section.

- The connection between Rb1 loss and sensitivity to navitoclax is promising and provocative yet is not thoroughly addressed:
 - o Rb1 loss should be confirmed or analyzed by western blot/IHC to in sensitive and resistant models.

Response: We had confirmed loss of RB1 by IHC in two of the navitoclax-responsive models in the previous manuscript (Figure 2B). In other cases the RB1 status is based on exome sequencing showing RB1 loss, and not based on IHC or western blotting. Therefore, as suggested by the reviewer, we have carried out IHC to further confirm RB1 loss in the two PDXs that we showed were responsive to navitoclax in vivo (BIDPC1 and BIDPC5). The results confirming RB1 loss are shown in a new figure (Supplementary Figure S1B). Interestingly, while BIDPC5 shows no staining, we do find weak nuclear staining in a minor subset of cells in BIDPC1. We indicate in the text that this may be nonspecific, but could possibly reflect retention of one copy of RB1 in a minor subclone. In either case, this would not alter the conclusions.

- o Similarly, in Figure 2C,D, no explanation is given on how “Rb1alt” was defined, on how the Genomics of Drug sensitivity database was analyzed. Critically, the plotted results labeled as “Rb1alt” seem to derived from mutational and not copy number analysis of the database, which is misleading. Indeed, there seems to be no significant association when RB1 copy number is used instead (cnaPANCAN389). These results should thus be re-interpreted and more experimental validation of the association to Rb1 loss should be provided, perhaps with other databases.

Response: The reviewer is correct and this should have been made clear in the original manuscript. In the Genomics of Drug Sensitivity in Cancer, selecting for effects in cells with mutations in specific genes includes both mutations and copy number alterations. This is what was shown in Fig. 2C, and we now clarify this point in the text. We have now analyzed separately the effect of RB1 copy number loss and found that it is similarly associated with increased sensitivity to navitoclax, WEHI, and ABT 737. This is shown in a new figure (Figure S2). It should also be noted that the analysis included only solid tumor derived cells as hematopoietic tumors are generally very dependent on BCL2.

- The functional link to Rb1 should also be experimentally reinforced. A single siRNA is used in a single cell line to substantiate this claim. More rigorous analyses using multiple siRNAs or rescue experiments and other cell line models should also be included.

Response: The reviewer is correct that the RB1 knockdown experiments were done in just one cell line (LNCaP). However, they were done both by transient expression of siRNA targeting RB1 and independently with a distinct stably expressed shRNA targeting RB1. In response to the reviewer’s suggestion we have now analyzed a breast cancer and a non small cell lung cancer (NSCLC) cell line with doxycycline-inducible RB1 shRNA knockdown (Figure S4). Notably, the RB1 knockdown results in a DNA damage response in both cell lines (see also comment below), but sensitizes to navitoclax only in the NCI-2030 (NSCLC) cells. The basis for the relative resistance in

MCF7 cells is not clear, but low levels of BAX and an increase in MCL-1 in response to RB1 knockdown may be factors (see also below) (Figure S9). We indicate in the text that more studies are needed to understand how genomic background modulates acute and chronic responses to RB1 downregulation or loss.

- Is a WT p53 status required for Rb1 loss-associated sensitivity to Navitoclax?

Response: This is a great question. Notably, related to the above point, NCI-2030 cells have a TP53 mutation, but we still see an increase in pRPA32 (indicating a DNA damage response) after RB1 knockdown (although less marked than in MCF7 cells). This is consistent with many other studies showing that ATR/ATM activation is not p53 dependent. In contrast, we showed that the downregulation of BIRC5/SURVIVIN in response to nucleotide pool disruption and subsequent DNA damage was mediated by p53. However, we also found that alternative mechanisms could sensitize to BH3 mimetics in TP53 deficient T47D cells. We did not in the previous manuscript go back and address whether the responses to single agent navitoclax in RB1 deficient cells were p53 dependent. However, this appears not to be the case as the BIDPC1 PDX, which had a robust response to navitoclax, is TP53 deficient. Moreover, as noted above, RB1 knockdown sensitized the NCI-2030 cells to navitoclax. We now make these points in the revised manuscript, and indicate that further studies are needed to determine whether RB1 loss sensitizes to BH3 mimetics by distinct mechanisms in TP53 intact versus deficient tumors.

- A major concern is that Rb1 loss is not mechanistically linked to replication stress and thus sensitize to Bcl-XI inhibition – the authors reference papers for which similar references could be found for other genetic alterations impacting genomic instability, other than Rb1. Indeed, loss of Tp53 (cnaPANCAN307) is significantly associated with sensitivity to navitoclax in the Genomics of Drug sensitivity database. The authors should study whether the same or different mechanisms of action (replication stress, stress response, etc) occur with Rb1 loss.
 - o How does Rb1 loss impact BH3 protein expression (ie, Bcl2, Bcl-XI, MCL1, etc)
 - o Does Rb1 loss lead to increased gH2aX/RPA32?

Response: The reviewer is correct that we did not pursue the mechanisms by which RB1 loss could sensitize to BH3 mimetic drugs. As suggested by the reviewer, we have now included data on effects of RB1 knockdown on DNA damage (gH2aX and pRPA32) (Figure S4) and on apoptosis related proteins (Figure S9). We show that RB1 knockdown in MCF7 (breast cancer) and NCI-2030 (NSCLC) cell lines causes a DNA damage response based on increased pRPA32 in NCI-2030 and in both pRPA32 and gH2Ax in MCF7 (Figure S4). We also show immunoblotting for MCL1, BCLXL, BCL2, BAX, BIM, and survivin in these cells pre- and post RB1 knockdown (Figure S9). Interestingly, RB1 knockdown in MCF7 cells resulted in increased levels of MCL1, BIM, and BAX, while survivin (which was expressed at low basal levels) was decreased. In contrast, survivin was expressed at high basal levels in NCI-H2030 cells and was modestly increased by RB1 knockdown, while there were no changes in the other apoptosis related proteins examined. Notably, NCI-H2030 cells have a G262V mutation in TP53 that may be a basis for their high basal survivin and no downregulation in response to RB1 loss. Based on this, we indicate that further studies are needed to address how effects of RB1 loss on apoptosis related genes and on responses to BH3 mimetics are modulated by other concurrent genomic alterations.

- The drug sensitivity screen is inadequately presented or explained. More detailed information should be provided for what the ICCB-L MoA library is, what drugs are included, results for different drug concentrations and, critically, how the statistical significance of the “hits” was assessed. If synergy was a criteria, how was this assessed? This is also important to assess the relative relevance of the different classes of drugs for overall sensitivity and to understand where navitoclax lies.

Response: The reviewer is correct that we did not go into great depth on the screen, as we focused on validating and further characterizing just one of the hits. We now have added more detailed information on how the screen was done and analyzed. With respect to the other drugs identified, we did note how they may be linked to navitoclax sensitivity by decreasing MCL1 or other mechanisms. We have expanded this discussion in the revised text.

- The authors show in vivo data to validate the efficacy of novel drug combinations. However, the statistical

presentation of the results are poor. Please show p values, n numbers and statistical tests used. Confirmation of induction of apoptosis in tumors (ie, by IHC stain) is warranted, at least at initial time points. Moreover, no toxicity studies are reported. The authors should at least document whether behavioral, body weight and hematological toxicities were observed.

Response: Figure 7 does confirm induction of apoptosis based on western blotting for cleaved caspase 3 (Figure 7C), and we also show a decrease in BIRC5/SURVIVIN in vivo in both models. However, the reviewer is correct regarding the statistical presentation, and we apologize that for not including this data. It has been added to the revised manuscript. With respect to toxicity, we did not note any apparent toxicity based on behavioral changes, and indicate in the revised text that the therapy did not alter body weight by more than 10%. Unfortunately we did not assess for hematological toxicities, but agree this would be a valuable addition and will be incorporated into future studies.

MINOR CONCERN

- Some results are not presented or explained clearly. Most CTG and CaspaseGlo assays are missing statistical analyses for significance. What sensitivity parameter is plotted on the heatmap of Figure 1B? Statistical significance is not assessed.

Response: We have done extensive revisions in the text in an effort to improve clarity and have added statistical analyses. We now also clarify in the text that the heatmap in Figure 1B is based on ranking of IC50 values across all the cells and treatments.